# HNDiff: Haze-Noise Diffusion for Image Dehazing

## Abstract

Existing diffusion-based methods have recently made significant progress in image dehazing. However, they typically neglect the physics of haze formation and reconstruct clean images from pure Gaussian noise, thereby limiting their restoration potential. To address this issue, we propose Haze-Noise Diffusion (HNDiff), a novel diffusion framework that embeds the atmospheric scattering model as an inductive bias. By grounding diffusion in physical principles, HNDiff ensures that the restoration aligns more closely with underlying mechanisms of haze formation. In its forward process, we introduce joint haze-noise diffusion with a haze-aware noise scheduler, which progressively adds both haze and noise to an image. Essentially, the scheduler adapts noise levels according to haze density, meaning that regions with heavier haze receive stronger noise injection to encourage content generation, while clearer regions receive lighter noise to better preserve details, which directly links the forward degradation process with the physics of haze. In the reverse process, we then derive a physically consistent dehazing-denoising process that simultaneously removes haze and noise to restore a clean image in a manner aligned with the forward degradation process. To further enhance practicality, we propose Latent HNDiff, which compiles clean latent priors that can be seamlessly integrated into existing dehazing networks to boost performance. Extensive experiments show that our work significantly improves leading dehazing backbones and achieves state-of-the-art results on benchmark datasets.

## 1 Introduction

Hazy weather conditions caused by atmospheric scattering frequently degrade image visibility by reducing contrast and obscuring scene details. Such degradation not only impairs human perception but also severely hinders the performance of many vision applications, such as object detection (Kim et al., 2024a; Wang et al., 2024a), semantic segmentation (Benigmim et al., 2024; Weber et al., 2024), and face recognition (Kim et al., 2024b; Mi et al., 2024). To address the challenges, single image dehazing has emerged as a feasible solution to restore a clear image from a single hazy input. However, such a task remains highly ill-posed due to the complex interplay of scattering coefficients, atmospheric light, and scene depth.

Driven by advances in deep learning, CNN-based methods (Dong et al., 2020; Wu et al., 2021; Bai et al., 2022; Cui et al., 2023) have achieved impressive results in image dehazing. Transformer-based approaches (Qin et al., 2020; Guo et al., 2022; Qiu et al., 2023; Cui et al., 2024; Fang et al., 2025) further improved performance by exploring long-range dependencies and global context. Recently, Mamba-based methods (Zheng & Wu, 2024; Li et al., 2025) have emerged as an efficient alternative with linear computational complexity. Despite the advancements, these methods still struggle in heavy haze scenarios, where most information is lost, leading to limited restoration quality.

In parallel, diffusion models (Ho et al., 2020; Rombach et al., 2022) have shown strong generative ability in image synthesis, producing results with rich details and sharp textures. Motivated by this progress, several studies (Yang et al., 2024; Wang et al., 2025) have applied diffusion algorithms to image dehazing. Yet, conventional diffusion models are fundamentally misaligned with the nature of haze. That is, they reconstruct clean images from pure Gaussian noise, and their stochastic nature (Ye et al., 2024) often causes deviations from the original image, thereby reducing restoration fidelity. More importantly, they usually neglect the physical properties of haze formation, resulting in suboptimal restoration performance. According to the Atmospheric Scattering

Figure 1: HNDiff leverages the ASM inductive bias, progressively adding haze and noise in the forward process and removing them in the reverse process for image dehazing.

Model (ASM) (Narasimhan & Nayar, 2003), a hazy image results from attenuated scene radiance and global atmospheric light. Haze density may vary spatially across the image, as it depends on both the light scattering effect and scene depth, and intensifies with the increasing scattering coefficient. Thus, haze exhibits structured, spatially varying degradations, unlike the random Gaussian noise typically assumed in conventional diffusion models.

Based on this observation, we present *Haze-Noise Diffusion* (HNDiff), a new framework that redefines the forward process through a haze-noise diffusion mechanism. Instead of injecting only Gaussian noise, we integrate the ASM into the diffusion process, mimicking the physical formation of haze by highlighting its spatially varying characteristics.

In the forward process, HNDiff carries out the haze-noise diffusion mechanism, which gradually adds both Gaussian noise and haze to a clean image, as illustrated in Figure 1. To better control this process, we introduce the haze-aware noise scheduler, which dynamically adjusts the noise level according to haze density: hazier regions are assigned higher noise to boost generative capacity, while clearer regions receive less noise to preserve detail fidelity. Progressive haze diffusion and adaptive noise scheduling require transmission maps from ASM, which are generally unavailable. To overcome this limitation, we consider the haze residual, defined as the incremental haze accumulated as the scattering coefficient increases. We develop a continuous accumulation formulation to represent this residual implicitly in HNDiff and thus eliminate the need for explicit transmission maps. Through this design, the forward process remains ingeniously consistent with ASM, enabling progressive haze addition in tandem with adaptive noise injection.

In the reverse process, we derive the dehazing-denoising process, which is grounded by the physical principles of ASM and can implicitly approximate noise and haze residuals through dedicated estimators, thereby removing haze and noise to restore clean images. However, directly applying diffusion in the image space incurs substantial computational overhead and may suffer from fidelity issues in severely degraded regions due to the stochastic nature of the diffusion process. To address these problems, we propose latent HNDiff, a prior generation network that integrates flexibly with dehazing backbones, allowing for more accurate and visually consistent restoration. This latent approach not only reduces computational cost but also enhances the applicability of the framework across diverse dehazing models.

The key contributions of our work are summarized as follows: First, we propose HNDiff, a novel diffusion-based framework that incorporates ASM as an inductive bias, specifically designed for image dehazing. Second, HNDiff implements a haze-noise diffusion process that adds both haze and noise in the forward pass, and a corresponding dehazing-denoising process with two dedicated estimators to respectively remove haze and noise in the backward pass. Third, we design the haze-aware noise scheduler to adaptively adjust noise levels based on hazy densities. Fourth, extensive experiments demonstrate that HNDiff consistently improves three representative dehazing models and achieves state-of-the-art performance on seven benchmark datasets.

## 2 RELATED WORK

### 2.1 IMAGE DEHAZING

**CNN-based Dehazing.** Deep learning has revolutionized image dehazing with CNN-based methods (Dong et al., 2020; Wu et al., 2021; Bai et al., 2022; Yang et al., 2022; Cui et al., 2023) achieving impressive breakthroughs. For instance, Dong et al. (2020) propose a boosted decoder combined with a dense feature fusion module to progressively restore haze-free images. Wu et al. (2021) introduce contrastive regularization within an autoencoder to learn from hazy and clear images for efficient dehazing. More recently, Cui et al. (2023) present a dual-domain selection mechanism and an efficient multi-scale network to further enhance restoration quality.

**Transformer-based Dehazing.** In addition to CNNs, Transformer-based methods have shown great promise in image dehazing by leveraging attention mechanisms to model long-range dependencies and global context (Qin et al., 2020; Guo et al., 2022; Song et al., 2023; Qiu et al., 2023; Valanarasu et al., 2022; Cui et al., 2024; Fang et al., 2025). For example, Qiu et al. (2023) approximate softmax-attention with a Taylor expansion to achieve linear complexity to reduce the computational overhead, complemented by multi-scale attention refinement for effective dehazing. Cui et al. (2024) design a multi-shape attention module with rectangle and dilated operations to enlarge receptive fields and boost performance. Fang et al. (2025) integrate phase and attention modules to leverage YCbCr textures for recovering clearer features in both frequency and spatial domains.

**Mamba-based Dehazing.** Mamba-based methods have recently emerged as efficient alternatives for image dehazing, capturing global context with linear computational complexity. Zheng & Wu (2024) combine convolution for local feature extraction with state space models to capture long-range dependencies in dehazing. Li et al. (2025) design an S-shaped stripe-based scanning strategy to better preserve locality and continuity, and use a channel-wise attention mechanism to aggregate sequences for more effective restoration.

**ASM-based Dehazing.** Beyond architectural advances, several studies (Shao et al., 2020; Chen et al., 2021; Yang et al., 2022; Wu et al., 2023; Fang et al., 2024; Shin et al., 2025) explicitly exploit the Atmospheric Scattering Model (ASM) to improve dehazing. Wu et al. (2023) design an ASM-based data generation pipeline to synthesize hazy images for training a VQGAN-based network. Fang et al. (2024) derive a cooperative unfolding network directly from ASM, jointly optimizing the transmission map and clean image. Shin et al. (2025) reformulate dehazing as an ASM-governed ODE flow, ensuring that the learned velocity field and transmission refinement remain consistent with the scattering physics.

Despite these advancements, most dehazing methods are still trained in an end-to-end regression manner that directly maps hazy inputs to clean outputs. Although some ASM-based approaches incorporate the Atmospheric Scattering Model as physical guidance to constrain this mapping, both regression-based and ASM-based methods still struggle under extremely dense haze, where severe information loss makes it difficult to recover realistic high-frequency details. In contrast, our method couples ASM with a diffusion process and leverages the generative capability of noise diffusion to compensate for missing content and restore plausible fine structures in heavily degraded regions.

## 2.2 DIFFUSION MODELS

**Diffusion for Low-level Vision.** Diffusion models (Ho et al., 2020; Rombach et al., 2022; Meng et al., 2022; Zhang et al., 2023; Wu et al., 2024) have shown strong generative capability in image synthesis and can produce results with rich details and sharp textures through forward noise diffusion and reverse denoising. This success has inspired much research exploring their potential in diffusion algorithms for various low-level vision tasks (Zhang et al., 2024; Garber & Tirer, 2024; Li et al., 2024; Liu et al., 2024a;b; Xia et al., 2023; Zheng et al., 2024; Rajagopalan et al., 2025; Luo et al., 2025; He et al., 2025). For example, Xia et al. (2023) employ diffusion models to extract compact priors used to guide a dynamic transformer for image recovery. Liu et al. (2024b) utilize a pre-trained diffusion model with task-specific priors for diverse image restoration tasks. Luo et al. (2025) present visual instruction-guided diffusion that models degradation patterns for all-in-one image restoration.

**Diffusion for Image Dehazing.** Within low-level vision, a number of studies have focused specifically on image dehazing using diffusion models (Yang et al., 2024; Wang et al., 2024b; Liu et al., 2024b; Rao et al., 2024; Wang et al., 2025; Liu et al., 2025; Zhou et al., 2025). For instance, Yang et al. (2024) exploit the semantic latent space of a pre-trained diffusion model to guide dehazing without retraining and iterative sampling. Wang et al. (2025) combine diffusion-based hazy image generation with accelerated fidelity-preserving sampling for efficient, high-quality dehazing. Liu et al. (2025) leverages diffusion models in the frequency domain with an amplitude residual encoder and a phase correction module to enhance unpaired image dehazing. Despite these advances, these methods rely on conventional noise diffusion initialized with pure Gaussian noise, which disregards the physical properties of haze formation. As a result, the stochastic nature (Ye et al., 2024) of the process often leads to deviations from the target restoration fidelity, often resulting in degraded performance and less consistent visual quality.

**Degradation-aware Diffusion.** Recently, a few studies have explored degradation-aware diffusion for image restoration. For example, Liu et al. (2024a) propose residual diffusion and operate

Figure 2: **Overview of Latent HNDiff.** The framework starts by using an image encoder to extract latent priors from a hazy input. These priors undergo a haze-noise diffusion process to produce the diffused hazy and noisy representation $Z_T$. During the reverse process, dehazing and denoising are performed jointly by iteratively estimating both noise and haze residuals to recover clean priors $Z_0$. Lastly, they are then integrated into a dehazing backbone through the Feature Gating Module (FGM) to improve the overall restoration quality.

on the difference between hazy and clean images, while Zhou et al. (2025) introduce a physics-guided dehazing diffusion by reformulating haze accumulation as a time-indexed process. However, these approaches either neglect the role of physical scene transmission in modeling haze degradation within the diffusion process or overlook haze density in the noise scheduling. To address these issues, our approach embeds ASM into the diffusion process as an inductive bias and adaptively adjusts noise addition according to haze density, achieving significantly improved dehazing performance.

## 3 METHOD

This section presents the proposed *Haze-Noise Diffusion* (HNDiff), a novel framework that integrates the Atmospheric Scattering Model (ASM)(Narasimhan & Nayar, 2003) into the diffusion process for image dehazing. As depicted in Figure 1, HNDiff defines a physics-guided forward *Haze-Noise Diffusion* process equipped with a *Haze-Aware Noise Scheduler* (HANS) and a reverse *Dehazing–Denoising Process*. In the forward direction, an input image is progressively degraded by jointly introducing haze and Gaussian noise through an increasing scattering coefficient, while HANS adaptively controls the noise level according to the local haze density so that the corruption follows ASM-guided haze formation. In the reverse direction, HNDiff starts from a hazy input perturbed by Gaussian noise and iteratively removes both haze and noise in a manner consistent with the ASM. For high-fidelity restoration and better efficiency, as illustrated in Figure 2, we further propose *Latent HNDiff*, where HNDiff serves as a prior generation network in the latent space and the learned prior is injected into a dehazing backbone via a Feature Gating Module (FGM), enabling plug-and-play enhancement of existing dehazing architectures. In the following, Section 3.1 details the Haze-Noise Diffusion process and HANS, Section 3.2 presents the Dehazing–Denoising Process, and Section 3.3 describes Latent HNDiff together with the FGM integration.

### 3.1 HAZE-NOISE DIFFUSION

Haze in image formation arises from atmospheric scattering, where scene radiance is attenuated during transmission and blended with global atmospheric light, leading to reduced visibility and a loss of detail. This process can be mathematically modeled by the ASM:

$$I_H(x) = I_0(x)\,\tau(x) + A\,(1 - \tau(x)), \text{ where } \tau(x) = e^{-\sigma(x)d(x)} \tag{1}$$

with $x$ denoting the pixel index. In (1), $I_H \in \mathbb{R}^{H \times W \times 3}$, $I_0 \in \mathbb{R}^{H \times W \times 3}$, $\tau \in \mathbb{R}^{H \times W \times 1}$, $A \in \mathbb{R}^3$, $\sigma \in \mathbb{R}^{H \times W \times 1}$, and $d \in \mathbb{R}^{H \times W \times 1}$ denote the hazy image, the clean scene radiance, the transmission map, the global atmospheric light, the scattering coefficient, and the scene depth, respectively. This formulation explicitly models haze formation, where larger scattering coefficients $\sigma$ or greater depths $d$ yield smaller transmission $\tau$, increasing haze density and reducing scene visibility. However, existing diffusion-based dehazing methods typically adopt conventional diffusion models that add zero-mean Gaussian noise and drive the image toward pure noise, which does not reflect the

structured, spatially varying haze described in (1). To bridge this gap, we introduce a *haze–noise diffusion* process in which ASM-based haze formation acts as a mean shift of the Gaussian corruption, so that the forward process follows a physically meaningful clean-to-hazy evolution while stochastic noise is injected around this trajectory.

**Forward Haze-Noise Diffusion.** In the forward process of HNDiff, we propose a haze-noise diffusion that embeds the physical haze formation process into noise diffusion. Specifically, a clean image is progressively degraded by both haze and Gaussian noise. The forward transition at time step $t$ is defined as

$$I_t(x) = I_{t-1}(x)e^{-\alpha_t\sigma(x)d(x)} + A\left(1 - e^{-\alpha_t\sigma(x)d(x)}\right) + \beta_t(x)\epsilon_t(x), \tag{2}$$

where $I_t$ denotes the intermediate hazy and noisy image at step $t$, $\alpha_t$ is the scaling factor, $\epsilon_t \sim \mathcal{N}(0, \mathbf{I})$ represents Gaussian noise, and $\beta_t(x)$ is the noise scaling coefficient at pixel $x$. As seen in (2), $I_{t-1}$ is further degraded according to (1), with scene radiance attenuated by the $\alpha_t$-scaled scattering coefficient, while noise is injected with another scaling coefficient $\beta_t(x)$.

**Haze-Aware Noise Scheduler.** Since haze density varies spatially, the generative capacity controlled by noise diffusion should also adapt across pixels. We therefore introduce a haze-aware noise scheduler, which defines the pixel-wise noise scaling coefficient as $\beta_t(x) = 1 - e^{-\alpha_t\sigma(x)d(x)}$. This design makes the injected noise explicitly dependent on haze density: pixels with heavier haze receive larger $\beta_t(x)$, thus introducing stronger noise that triggers diffusion to reconstruct severely degraded details; conversely, pixels with lighter haze yield smaller $\beta_t(x)$, injecting less noise to preserve content fidelity through regression. This adaptive scheduling enables the forward process to jointly model haze degradation and stochastic corruption.

**Sampling Probability and Reparameterization.** Inspired by Liu et al. (2024a), we regard degradation in the forward process as a deterministic mean shift. From (2), each step from $I_{t-1}$ to $I_t$ can thus be expressed as a Gaussian transition, where the mean is shifted by haze and stochastic perturbations are introduced by Gaussian noise:

$$q(I_t(x) \mid I_{t-1}(x), \phi) := \mathcal{N}\left(I_t(x) \,\Big|\, I_{t-1}(x)e^{-\alpha_t\sigma(x)d(x)} + A\left(1 - e^{-\alpha_t\sigma(x)d(x)}\right), \beta_t^2(x)\right), \tag{3}$$

where $\phi = \{d(x), \sigma(x), A\}$. By iterating (3), we obtain a sequence of progressively hazy and noisy images $\{I_1, I_2, \ldots, I_T\}$ through a $T$-step diffusion process, with the complete forward sampling probability $q(I_{1:T}(x) \mid I_0(x), \phi) = \prod_{t=1}^{T} q(I_t(x) \mid I_{t-1}(x), \phi)$. However, existing dehazing datasets provide only hazy-clean image pairs and do not include $\phi$ (i.e., atmospheric light, scattering coefficients, and scene depth necessary to compute transmission).

To address this limitation, we apply the reparameterization trick (Ho et al., 2020) to (3) and obtain the conditional distribution after $T$ steps as

$$q(I_T(x) \mid I_0(x), \phi) = \mathcal{N}\left(I_T(x) \,\Big|\, I_0(x)e^{-\sum_{t=1}^{T}\alpha_t\sigma(x)d(x)} + A\left(1 - e^{-\sum_{t=1}^{T}\alpha_t\sigma(x)d(x)}\right), \bar{\beta}_T^2(x)\right), \tag{4}$$

where $\alpha_t = \frac{1}{T}$, $\forall t \in \{1, 2, \ldots, T\}$, and $\bar{\beta}_T(x) = \sqrt{\frac{(1-e^{-(1/T)\sigma(x)d(x)})(1-e^{-2\sigma(x)d(x)})}{1+e^{-(1/T)\sigma(x)d(x)}}}$. The complete derivation of (4) is provided in Appendix A.1. It follows that the hazy and noisy image $I_T$ can be sampled from $q(I_T \mid I_0)$ via

$$I_T(x) = I_0(x)e^{-\sigma(x)d(x)} + A\left(1 - e^{-\sigma(x)d(x)}\right) + \bar{\beta}_T(x)\epsilon(x) = I_H(x) + \bar{\beta}_T(x)\epsilon(x), \tag{5}$$

where $I_T$ is generated in a single step by injecting noise into the hazy image $I_H$ via the haze-aware noise scheduler. This formulation preserves the Gaussian nature of the diffusion process while embedding ASM directly into the mean of the distribution through a physically grounded shift. As $\bar{\beta}_T(x)$ still relies on the transmission map, we introduce a learnable haze estimator to implicitly approximate it. The optimization details for the haze estimator are provided in Section 3.3.

## 3.2 DEHAZING-DENOISING PROCESS

In the reverse generation procedure, we aim to progressively remove both haze and noise from the degraded observation $I_T$ to recover the clean image $I_0$. Unlike conventional diffusion models that start from pure Gaussian noise, our method initializes from the hazy-noisy sample $I_T$ drawn from

the Gaussian distribution (4). Inspired by the deterministic sampling formulation in Song et al. (2021), we define the reverse transition distribution as

$$p_\theta(I_{t-1}(x) \mid I_t(x)) = q_\delta(I_{t-1}(x) \mid I_t(x), I_0(x), \phi). \tag{6}$$

The transition probability $q_\delta$ in (6) is defined as

$$q_\delta(I_{t-1}(x) \mid I_t(x), I_0(x), \phi) = \mathcal{N}\left(I_{t-1}(x) \mid \mu_t(x), \delta_t^2(x)\right), \quad \text{where} \tag{7}$$

$$\mu_t(x) = I_0(x)e^{-\sum_{s=1}^{t-1}\alpha_s\sigma(x)d(x)} + A\left(1 - e^{-\sum_{s=1}^{t-1}\alpha_s\sigma(x)d(x)}\right) + \sqrt{\bar{\beta}_{t-1}^2(x) - \delta_t^2(x)}\epsilon_{t-1}(x), \tag{8}$$

and $\delta_t^2 = \eta \cdot \frac{\beta_t^2 \bar{\beta}_{t-1}^2}{\bar{\beta}_t^2}$ is a variance term that controls sampling stochasticity. When $\eta = 0$, this yields a deterministic sampling. From 4, we can derive

$$I_0(x) = (I_t - (1 - e^{-\sum_{s=1}^{t-1}\alpha_s\sigma(x)d(x)})A - \bar{\beta}_t\epsilon_t)e^{\sum_{s=1}^{t-1}\alpha_s\sigma(x)d(x)}. \tag{9}$$

By substituting (9) into (7) and simplifying, we obtain the sampling equation for $I_{t-1}(x)$ as

$$I_{t-1}(x) = \left(I_t(x) - N_t(x)\left(1 - e^{-\alpha_t\sigma(x)d(x)}\right)\right)e^{\alpha_t\sigma(x)d(x)}, \tag{10}$$

where $N_t(x) = A + \epsilon_t(x)$ denotes the atmospheric noise, which is composed of the atmospheric light term and a Gaussian noise term. To reconstruct $I_0$, we iterate (10) with two learnable estimators. One is the noise estimator $N_t^\theta(I_t, I_H, t)$, which approximates $N_t$. The other is the haze estimator $1 - e^{-\alpha_t o^\theta(I_t, I_H, t)}$, which approximates the residual transmission term $1 - e^{-\alpha_t\sigma d}$ (the complement of the transmission), where $o^\theta(I_t, I_H, t)$ is a learnable network estimating the scattering–depth product $\sigma d$. Here, we omit $A$ for simplicity, as it can be incorporated separately into the haze reconstruction. Complete derivations of the variational lower bound and sampling formulation are provided in Appendix A.2, A.3. In the following, we detail the optimization of the haze estimator and noise estimators in the latent space.

### 3.3 LATENT HNDIFF

Performing diffusion-based restoration directly in image space, as noted in Rombach et al. (2022); Chen et al. (2023), incurs substantial computational overhead, fidelity degradation, and slower, less stable convergence. To address these challenges, and inspired by prior works (Rombach et al., 2022; Chen et al., 2023; Xia et al., 2023), we present **Latent HNDiff**. As illustrated in Figure 2, Latent HNDiff applies HNDiff in the latent space and serves as a prior generator to enhance dehazing through a three-stage training strategy. By embedding physically grounded haze formation into the latent diffusion process, Latent HNDiff encourages latent features to encode haze-aware information, thereby capturing meaningful physical representations.

**Stage 1: Ground-truth Prior Pretraining.** We first pretrain a dehazing network equipped with an Image Encoder (IE) and a Feature Gating Module (FGM). Given a hazy image $I_H$ and its clean counterpart $I_0$, we concatenate them and feed the result into the IE to extract the ground-truth prior $Z_{gt} = \text{IE}(\text{Concat}(I_H, I_0)) \in \mathbb{R}^{\frac{H}{4} \times \frac{W}{4} \times 4}$. The prior $Z_{gt}$ is fused with encoder and decoder features $F_i^{in} \in \mathbb{R}^{h_i \times w_i \times c_i}$ at each scale $i$ of the dehazing network through the FGM, producing the fused features $F_i^{out}$. Within FGM, $Z_{gt}$ is first passed through a shared-weight Prior Encoder (PE) to obtain a compact representation, which is then linearly projected to generate modulation parameters that adaptively modulate the input features:

$$z_{gt} = \text{PE}(Z_{gt}) = \text{MLP}(\text{AvgPool2D}(\text{Unshuffle}(Z_{gt}))) \in \mathbb{R}^{1 \times C} \text{ and} \tag{11}$$

$$F_i^{out} = F_i^{in} \times z^{\alpha_i} + z^{\beta_i}, \text{ where } (z^{\alpha_i} \in \mathbb{R}^{1 \times c_i}, z^{\beta_i} \in \mathbb{R}^{1 \times c_i}) = \text{Linear}(z_{gt}). \tag{12}$$

$C$ in (11) denotes the channel dimension of the projected prior vector, while $c_i$ in (12) represents the number of feature channels at the $i$-th scale of the dehazing network. The fused features across all scales are subsequently decoded to yield the dehazed image $I_{dehz}$, which is supervised by the clean reference $I_0$, ensuring that the network effectively learns to exploit the ground-truth prior $Z_{gt}$.

**Stage 2: Latent HNDiff Optimization.** We estimate the ground-truth prior $Z_{gt}$ from $I_H$ using HNDiff in the absence of the clean counterpart $I_0$. Specifically, a second IE extracts $Z_H \in \mathbb{R}^{\frac{H}{4} \times \frac{W}{4} \times 4}$ from $I_H$. We then apply haze-noise diffusion (5) to $Z_H$ using the haze estimator and haze-aware noise scheduler, obtaining a degraded latent $Z_T$. Next, the dehazing-denoising process

Table 1: Quantitative results on six benchmark datasets. Values in parentheses represent the improvements of HNDiff over the corresponding baselines.

| Model | NH-HAZE | | O-HAZE | | Dense-HAZE | | RW$^2$AH | | SOTS-Indoor | | SOTS-Outdoor | |
|---|---|---|---|---|---|---|---|---|---|---|---|---|
| | PSNR | SSIM | PSNR | SSIM | PSNR | SSIM | PSNR | SSIM | PSNR | SSIM | PSNR | SSIM |
| MSBDN | 17.97 | 0.659 | 24.36 | 0.749 | 15.13 | 0.555 | 21.51 | 0.595 | 33.67 | 0.985 | 33.48 | 0.982 |
| FFA-Net | 18.13 | 0.647 | 22.12 | 0.770 | 15.70 | 0.549 | 18.73 | 0.556 | 36.39 | 0.989 | 33.57 | 0.984 |
| Dehamer | 20.66 | 0.684 | 25.11 | 0.777 | 16.62 | 0.560 | 20.84 | 0.581 | 36.63 | 0.988 | 35.18 | 0.986 |
| MB-TaylorFormer | 20.43 | 0.688 | 25.05 | 0.788 | 16.66 | 0.560 | 21.37 | 0.608 | 40.71 | 0.992 | 37.42 | 0.989 |
| FocalNet | 20.36 | 0.696 | 25.46 | 0.791 | 16.95 | 0.597 | 21.93 | 0.635 | 40.82 | 0.992 | 37.71 | 0.995 |
| ConvIR | 20.65 | 0.692 | 25.25 | 0.784 | 16.86 | 0.600 | 21.99 | 0.640 | 41.53 | 0.994 | 37.95 | 0.994 |
| SGDN | 20.13 | 0.680 | 24.59 | 0.778 | 16.60 | 0.571 | 22.24 | 0.631 | 41.01 | 0.992 | 36.22 | 0.986 |
| HNDiff (FocalNet) | 20.89 | 0.697 | 26.32 | 0.801 | 17.29 | 0.599 | 22.29 | 0.647 | 41.19 | 0.994 | 38.10 | 0.996 |
| | (+0.53) | (+0.001) | (+0.86) | (+0.010) | (+0.34) | (+0.002) | (+0.36) | (+0.012) | (+0.37) | (+0.002) | (+0.39) | (+0.001) |
| HNDiff (ConvIR) | 21.23 | 0.701 | 26.20 | 0.799 | 17.18 | 0.623 | 22.25 | 0.646 | 42.10 | 0.995 | 38.83 | 0.995 |
| | (+0.58) | (+0.009) | (+0.95) | (+0.015) | (+0.32) | (+0.023) | (+0.26) | (+0.006) | (+0.57) | (+0.001) | (+0.88) | (+0.001) |
| HNDiff (SGDN) | 20.64 | 0.686 | 25.40 | 0.782 | 17.17 | 0.611 | 22.81 | 0.653 | 41.47 | 0.995 | 37.10 | 0.991 |
| | (+0.51) | (+0.006) | (+0.81) | (+0.004) | (+0.57) | (+0.040) | (+0.57) | (+0.022) | (+0.46) | (+0.003) | (+0.88) | (+0.005) |
| Avg Gains | **+0.54** | **+0.005** | **+0.87** | **+0.010** | **+0.41** | **+0.022** | **+0.40** | **+0.013** | **+0.47** | **+0.002** | **+0.72** | **+0.002** |

(10) iteratively removes haze and noise, producing the refined prior $Z_0$ as an estimate of $Z_{gt}$. Previous diffusion-based approaches (Chen et al., 2023; Xia et al., 2023; Salimans & Ho, 2022; Rao et al., 2024) typically impose supervision only on the final reconstructed output, allowing gradients to propagate backward through the entire diffusion trajectory and thereby amortizing step-wise supervision. Inspired by this idea, we design a trajectory-level supervision scheme in the latent space. Specifically, we define a latent-prior loss as $\mathcal{L}_{prior} = \|Z_0 - Z_{gt}\|_1$, where $Z_0$ is reconstructed from $Z_T$ by recursively applying the shared haze and noise estimators. This design enforces consistency between the reconstructed latent representation and the ground-truth prior across the entire diffusion process. Further details are provided in Appendix A.2.

**Stage 3: Joint Fine-tuning.**    At last, we jointly optimize the pretrained IE, HNDiff, the FGM, and the dehazing backbone. The dehazed image $I_{dehz}$ is reconstructed by integrating the learned prior $Z_0$ and is supervised with $I_0$ using the standard loss function of the dehazing backbone. This stage ensures that the learned diffusion prior $Z_0$ consistently enhances dehazing performance.

## 4 EXPERIMENTS

### 4.1 EXPERIMENTAL SETUP

**Implementation Details.**    HNDiff is composed of four key components: the Image Encoder (IE), the Feature Gating Module (FGM), the Haze Estimator, and the Noise Estimator. The IE consists of six residual blocks and four CNN layers, while the FGM is implemented with a pooling operation and a lightweight MLP. Both the Haze Estimator and Noise Estimator share the same network architecture, which is a simplified U-Net (Liu et al., 2024a). In practice, we set the diffusion step to $T = 4$. The overall framework (Stage 3) is optimized with the default hyperparameters of each dehazing backbone (e.g., learning rate, number of epochs, batch size, and optimizer) to ensure fair comparisons.

**Dehazing Models and Datasets.**    We adopt three state-of-the-art image dehazing models, including FocalNet (Cui et al., 2023), ConvIR (Cui et al., 2024), SGDN (Fang et al., 2025) to validate the effectiveness of HNDiff. Following prior studies, we conduct experiments on one widely used synthetic dataset, SOTS-Indoor and SOTS-Outdoor (Li et al., 2018), and four real-world benchmarks: NH-HAZE (Ancuti et al., 2021), O-HAZE (Ancuti et al., 2018), Dense-HAZE (Ancuti et al., 2019), and RW$^2$AH (Fang et al., 2025). The SOTS-Indoor dataset consists of 13,990 training pairs and 500 testing pairs. The SOTS-Outdoor dataset consists of 313,950 training pairs and 500 testing pairs. Both NH-HAZE and Dense-HAZE provide 50 training pairs and 5 testing pairs. O-HAZE offers 40 training pairs and 5 testing pairs. RW$^2$AH is a real-world hazy dataset that includes 1,406 training pairs and 352 testing pairs.

### 4.2 EXPERIMENTAL RESULTS

**Quantitative Results.**    As shown in Table 1, we compare the dehazing performance of state-of-the-art methods and their HNDiff-enhanced versions, where the values in parentheses indicate the improvements made by HNDiff over the corresponding dehazing baselines. The results clearly demonstrate that HNDiff consistently enhances the performance of each baseline and outperforms

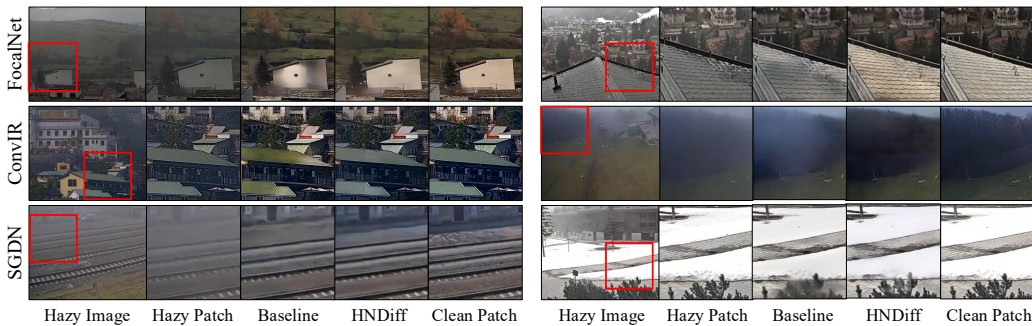

Figure 3: Qualitative results on the RW$^2$AH dataset.

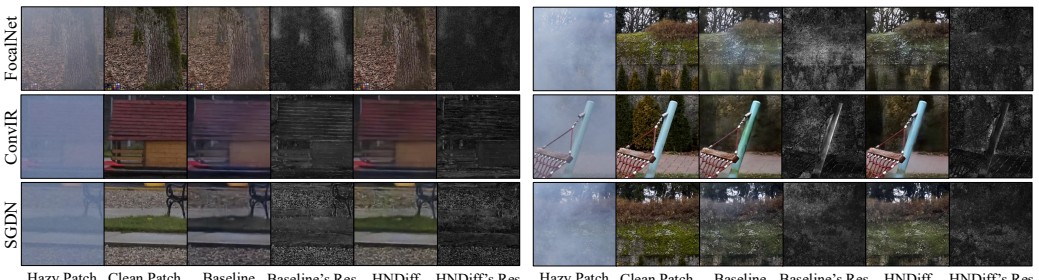

Figure 4: Qualitative results on the O-HAZE (left) and NH-HAZE (right) datasets. "Res" denotes residual maps between outputs and ground truth, where darker intensities indicate smaller errors.

previous state-of-the-art methods. Specifically, HNDiff yields average PSNR/SSIM improvements of +0.54/+0.005, +0.87/+0.010, +0.41/+0.022, +0.40/+0.013, +0.47/+0.002, and +0.72/0.002 on the NH-HAZE, O-HAZE, Dense-HAZE, RW$^2$AH, SOTS-Indoor, and SOTS-Outdoor test sets, respectively. Additionally, HNDiff achieves average PSNR/SSIM improvements of +0.48/+0.005, +0.59/+0.009, and +0.63/+0.013 on baselines FocalNet, ConvIR, and SGDN, respectively. Overall, HNDiff delivers an average gain of +0.57 PSNR and +0.009 SSIM across all datasets and baselines, highlighting its strong generalization ability and effectiveness as a prior generation network for image dehazing.

**Qualitative Results.** We present qualitative comparisons between three baseline models and their HNDiff-enhanced counterparts. Figure 3 presents the results on the RW$^2$AH test set, while Figure 4 shows the results on the NH-HAZE and O-HAZE test sets, including an additional "Res" column for better comparison. The residual maps are obtained by subtracting the ground truth from the model outputs, where lower intensities indicate smaller errors and thus higher reconstruction quality. As shown, HNDiff consistently produces cleaner and more visually compelling dehazed images compared to the baselines. By integrating HNDiff into the latent space of the dehazing networks, we exploit its capacity to model rich and realistic image priors while preserving fidelity to the underlying clean image structures. More qualitative results are provided in Appendix A.6.

### 4.3 ABLATION STUDIES

To assess the contributions of the proposed components in HNDiff, we conduct a series of ablation studies using FocalNet as the baseline dehazing model. Specifically, we evaluate the effectiveness of each component, compare the prior generator with different diffusion mechanisms, examine HNDiff against the baseline under equivalent parameter counts, investigate the impact of the three-stage training strategy, analyze the effect of varying the number of diffusion steps, inspect the haze residual modeling in latent space, and compare the dehazing results of applying HNDiff in image space and latent space. All experiments are conducted using the default training configuration of FocalNet and evaluated on the NH-HAZE test set.

**Effectiveness of Each Component.** Our ablation study, detailed in Table 2, evaluates the contribution of each component in HNDiff. *Net1* denotes the baseline dehazing model. *Net2* represents a conventional DDPM-based variant that employs only noise diffusion, while *Net3* is a variant that incorporates only haze diffusion and omits noise diffusion. *Net4* adopts both noise and haze diffusion but excludes the haze-aware noise scheduler (HANS). Finally, *Net5* is the complete HNDiff design.

Table 2: Ablation study on the effectiveness of noise diffusion, haze diffusion, and HANS.

| Model | Noise Diffusion | Haze Diffusion | HANS | PSNR (dB) |
|---|---|---|---|---|
| Net1 | | | | 20.36 (baseline) |
| Net2 | ✓ | | | 20.46 |
| Net3 | | ✓ | | 20.61 |
| Net4 | ✓ | ✓ | | 20.68 |
| Net5 | ✓ | ✓ | ✓ | **20.89** |

Table 3: Ablation study on the effectiveness of the three-stage training strategy.

| Model | Stage 1 | Stage 2 | Stage 3 | PSNR (dB) |
|---|---|---|---|---|
| Net1 | | | | 20.36 (baseline) |
| Net2 | ✓ | | | 21.36 (upper bound) |
| Net3 | | | ✓ | 20.51 |
| Net4 | ✓ | ✓ | | 20.58 |
| Net5 | ✓ | ✓ | ✓ | **20.89** |

Table 4: Comparison of different prior generators, including U-Net and three diffusion mechanisms.

| Model | Prior Generator | PSNR (dB) | SSIM |
|---|---|---|---|
| Net1 | N/A | 20.36 | 0.696 |
| Net2 | U-Net | 20.41 | 0.696 |
| Net3 | DDPM | 20.46 | 0.695 |
| Net4 | RDDM | 20.43 | 0.690 |
| Net5 | HNDiff | **20.89** | **0.697** |

Table 5: Comparison between HNDiff and baseline variants with comparable parameter counts.

| | FocalNet | FocalNet$^+$ | FocalNet$^*$ | HNDiff |
|---|---|---|---|---|
| Params (M) | 3.74 | 8.40 | 8.28 | 7.82 |
| FLOPs (G) | 30.53 | 68.54 | 64.05 | 36.38 |
| PSNR (dB) | 20.36 | 20.37 | 20.51 | **20.89** |

Table 6: Analysis of diffusion step setting.

| Time step | 0 | 2 | 4 | 6 | 8 |
|---|---|---|---|---|---|
| PSNR (dB) | 20.36 | 20.69 | **20.89** | 20.84 | 20.81 |

The results show that both *Net2* and *Net3* surpass the baseline, demonstrating the individual benefits of noise and haze diffusion. Moreover, *Net5* achieves the best performance, indicating that the joint integration of both diffusion processes together with HANS provides complementary gains. These findings highlight the importance of incorporating haze-aware design in order to enhance dehazing effectiveness.

**Effectiveness of Three-stage Training Strategy.** We evaluate the effectiveness of the three-stage training strategy, as shown in Table 3. *Net1* represents the baseline dehazing model. *Net2* serves as the upper bound using ground-truth prior $Z_{gt}$. *Net3* corresponds to optimizing HNDiff jointly with the dehazing model without Stage 1 and Stage 2 pre-training, thus serving as a purely data-driven baseline. *Net4* is the model obtained with Stage 1 and Stage 2 pre-training but without Stage 3 joint fine-tuning. Finally, *Net5* adopts the complete three-stage training and achieves the best performance. These results clearly demonstrate the effectiveness of the three-stage training strategy in exploiting the complementary benefits of pre-training and joint optimization.

**Comparison of Prior Generators with Different Diffusion Mechanisms.** Table 4 evaluates the baseline dehazing model augmented with different prior generation methods, including U-Net, DDPM (Ho et al., 2020), RDDM (Liu et al., 2024a), and our proposed HNDiff. *Net1* denotes the baseline model without a prior generator. *Net2* employs a U-Net to generate priors directly, without any diffusion process. *Net3*, *Net4*, and *Net5* are the dehazing models enhanced with priors generated by DDPM, RDDM, and HNDiff, respectively. Although integrating the standard diffusion process (*Net3*) or the residual diffusion process (*Net4*) improves performance over the baseline, the gain is just comparable to that of *Net2*, which uses a U-Net without diffusion. In contrast, HNDiff explicitly embeds the atmospheric scattering model into the diffusion process, yielding consistent and superior improvements compared to both standard and residual diffusion mechanisms. We present the dehazed results of different diffusion mechanisms in Appendix A.5.

**Comparison of HNDiff and Baselines with Equivalent Parameter Counts.** To ensure a fair comparison under similar parameter budgets, we evaluate HNDiff against enlarged baseline variants, as reported in Table 5. The *FocalNet$^+$* variant increases the base channel size from 32 to 48, while another variant *FocalNet$^*$* expands the number of residual blocks from 4 to 10. Although both variants substantially increase model complexity in terms of parameters and FLOPs, they yield only marginal PSNR gains over the baseline FocalNet. In contrast, HNDiff achieves the best performance of 20.89dB in PSNR with a lower parameter count (7.82M) and significantly reduced FLOPs (36.38G). These results demonstrate that integrating the proposed diffusion prior is more effective than simply scaling the network capacity.

**Analysis of Diffusion Step Setting.** Table 6 analyzes the impact of varying diffusion steps $T$ (0–8). Without diffusion guidance ($T = 0$), performance is limited to 20.36dB PSNR. Increasing $T$ improves results, peaking at 20.89dB with $T = 4$, while larger $T$ brings no further gains. These results indicate that our model converges effectively with only four diffusion steps, demonstrating that large numbers of diffusion iterations are unnecessary for achieving strong performance.

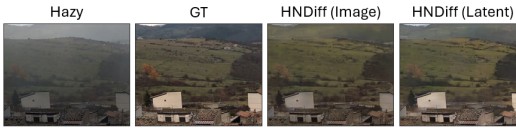

Figure 5: Visualization of latent representations across reverse diffusion steps.

Figure 6: Qualitative results of applying HNDiff in image space and latent space.

Table 7: Comparison of applying HNDiff in image space and latent space on the $RW^2AH$ dataset.

| Metric | FocalNet | HNDiff (Image) | HNDiff (Latent) |
|---|---|---|---|
| PSNR (dB) | 21.18 | 21.37 | **21.52** |
| SSIM | 0.5970 | 0.6166 | **0.6254** |
| FLOPs (G) | 30.53 | 65.59 | 36.38 |

**Analysis of Haze Residual Modeling in Latent Space.** To verify that HNDiff models haze formation in latent space, we analyze diffusion prior outputs across reverse steps. Although the model is trained with $T = 4$ steps, we examine intermediate latent representations by performing $t \in [0, 1, 2, 3, 4]$ reverse steps starting from the fully hazy latent $Z_4$, resulting in the sequence $[Z_4, Z_3, \ldots, Z_0]$. For visualization, we compute the channel-wise mean of each latent and downsample $I_H$ to the same spatial resolution only for visualization. As shown in Figure 5, the representations progressively transition from hazy ($Z_4$) to clean ($Z_0$), confirming that HNDiff captures a progressive hazy-to-clean structure in the latent space and enables interpretable modeling.

**Comparison of Applying HNDiff in Image Space and Latent Space.** Table 7 compares an image-space variant, *HNDiff (Image)*, which applies our haze-noise diffusion directly to RGB images, and a latent-space variant, *HNDiff (Latent)*, which operates in the latent space of a FocalNet on the real-world $RW^2AH$ dataset. The two variants use the same U-Net as the haze/noise estimator and both improve over the baseline, while *HNDiff (Latent)* achieves the best performance (21.52 dB PSNR, 0.6254 SSIM) with only minimal additional FLOPs overhead (36.38G), confirming its advantage in content fidelity. Figure 6 further shows that *HNDiff (Latent)* produces results closer to the ground truth, supporting our choice of the latent formulation in the main experiments.

## 5 LIMITATIONS

HNDiff is tailored to the Atmospheric Scattering Model and thus cannot be directly applied to other degradations such as motion blur, raindrops, or low-light conditions. Extending it to these scenarios requires integrating degradation-specific priors (e.g., object motion, rain masks, exposure time), which we leave as future work.

## 6 CONCLUSION

We propose Haze-Noise Diffusion (HNDiff), a novel diffusion-based framework for image dehazing. HNDiff integrates the atmospheric scattering model into the diffusion framework, jointly performing haze diffusion and noise diffusion to account for the physical properties of haze formation. In the forward process, HNDiff progressively degrades a clean image by introducing both haze and noise through a haze-noise diffusion, with a haze-aware noise scheduler that adaptively adjusts noise levels according to haze density. In the reverse process, HNDiff restores the image by removing both haze and noise through its dehazing-denoising process. To enhance the existing dehazing methods, we incorporate HNDiff into their latent spaces as a prior generator, seamlessly integrating the learned prior into each encoder/decoder block via our proposed Feature Gating Module to generate higher-quality dehazed results. Extensive experimental results have demonstrated that our method effectively improves the performance of three state-of-the-art dehazing models across seven dehazing datasets.

## ETHICS STATEMENT

This work focuses on designing a diffusion-based model for single-image dehazing. It does not involve human subjects, personal data, or sensitive content, and it follows the ICLR Code of Ethics.

All experiments are conducted on publicly available dehazing datasets with appropriate licenses. We do not anticipate any privacy, safety, or fairness concerns, and our method is intended solely to improve image quality in adverse weather conditions without harmful applications.

## REPRODUCIBILITY STATEMENT

Detailed model architecture (Section 3.3), training settings and dataset preparation (Section 4.1), and complete proofs (Appendix A) are provided to ensure reproducibility. The full codebase and pretrained weights will be released publicly upon acceptance.

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

# A APPENDIX

## A.1 ONE-STEP DIFFUSION DERIVATION FOR HNDIFF

Real-world dehazing datasets provide only hazy–clean image pairs, without the transmission maps and atmospheric light required in (1). To address this limitation, we adopt the reparameterization trick (Ho et al., 2020) to derive a one-step diffusion formulation suitable for real-world scenarios.

For clarity of notation, we omit the spatial dependency $(x)$ in the following derivation, since all operations are defined point-wise in the image domain.

In the haze–noise diffusion process of HNDiff, the forward process is defined as

$$q(I_t \mid I_{t-1}, \phi) := \mathcal{N}\big(I_t \mid I_{t-1}e^{-\alpha_t \sigma d} + A\big(1 - e^{-\alpha_t \sigma d}\big),\, \beta_t^2\big), \tag{13}$$

$$q(I_{1:T} \mid I_0, \phi) = \prod_{t=1}^{T} q(I_t \mid I_{t-1}, \phi), \quad \text{where} \quad \phi = \{d, \sigma, A\}. \tag{14}$$

Let $\tau_t = e^{-\alpha_t \sigma d}$ denote the per-step transmission rate. Then (13) can be rewritten as

$$I_t = \tau_t I_{t-1} + (1 - \tau_t)A + \beta_t \epsilon_t, \qquad \epsilon_t \sim \mathcal{N}(0, 1). \tag{15}$$

By recursively expanding from the last step, we obtain

$$
\begin{aligned}
I_T &= \tau_T I_{T-1} + (1 - \tau_T)A + \beta_T \epsilon_T \\
&= \tau_T[\tau_{T-1}I_{T-2} + (1 - \tau_{T-1})A + \beta_{T-1}\epsilon_{T-1}] + (1 - \tau_T)A + \beta_T \epsilon_T \\
&= (\tau_T \tau_{T-1})I_{T-2} + (1 - \tau_T \tau_{T-1})A + \tau_T \beta_{T-1}\epsilon_{T-1} + \beta_T \epsilon_T \\
&\vdots \\
&= \Big(\prod_{s=1}^{T} \tau_s\Big)I_0 + \Big(1 - \prod_{s=1}^{T} \tau_s\Big)A + \sum_{t=1}^{T} \Big(\prod_{s=t+1}^{T} \tau_s\Big)\beta_t \epsilon_t.
\end{aligned}
\tag{16}
$$

Define the cumulative transmission as

$$\bar{\tau}_T = \prod_{s=1}^{T} \tau_s = \exp\Big(-\sigma d \sum_{s=1}^{T} \alpha_s\Big), \tag{17}$$

and the aggregated noise variance as

$$\bar{\beta}_T^2 = \sum_{t=1}^{T} \bigg(\prod_{s=t+1}^{T} \tau_s^2\bigg) \beta_t^2. \tag{18}$$

Since $\{\epsilon_t\}$ are i.i.d. standard Gaussian variables, their weighted sum remains Gaussian. Thus,

$$\sum_{t=1}^{T} \Big(\prod_{s=t+1}^{T} \tau_s\Big)\beta_t \epsilon_t = \bar{\beta}_T \epsilon, \qquad \epsilon \sim \mathcal{N}(0, 1). \tag{19}$$

Consequently, the full forward process simplifies to a *one-step* form:

$$I_T = \bar{\tau}_T I_0 + (1 - \bar{\tau}_T)A + \bar{\beta}_T \epsilon, \tag{20}$$

with the corresponding marginal distribution

$$q(I_T \mid I_0, \phi) = \mathcal{N}\Big(I_T \,\Big|\, \bar{\tau}_T I_0 + (1 - \bar{\tau}_T)A,\, \bar{\beta}_T^2\Big). \tag{21}$$

Finally, by setting $\alpha_t = \frac{1}{T}$ for all $t$, the cumulative transmission reduces to

$$\bar{\tau}_T = e^{-\sigma d},$$

and the aggregated noise variance becomes

$$\bar{\beta}_T = \sqrt{\frac{(1 - e^{-(1/T)\sigma d})(1 - e^{-2\sigma d})}{1 + e^{-(1/T)\sigma d}}}.$$

Thus, $I_T$ can be expressed as

$$I_T = e^{-\sigma d}I_0 + (1 - e^{-\sigma d})A + \bar{\beta}_T \epsilon = I_H + \bar{\beta}_T \epsilon,$$

where $I_H = e^{-\sigma d}I_0 + (1 - e^{-\sigma d})A$ corresponds to the hazy image defined by the Atmospheric Scattering Model (1). This shows that the final forward step $I_T$ can be obtained by directly adding Gaussian noise to the hazy image. The one-step formulation is mathematically equivalent to the full forward process, while providing a more computationally efficient approximation that jointly captures haze formation and noise accumulation in a single Gaussian transition.

## A.2 ELBO AND OPTIMIZATION FOR HNDIFF

For clarity of notation, we omit the spatial dependency $(x)$ in the following derivation, since all operations are defined point-wise in the image domain.

To reconstruct the clean image $I_0$ from the degraded observation $I_T$, we adopt the variational inference framework of DDPM (Ho et al., 2020) and derive an evidence lower bound (ELBO) that explicitly incorporates the physical parameters $\phi = \{d, \sigma, A\}$. The joint ELBO is given by

$$\log p_\theta(I_0) \geq \mathbb{E}_{q(I_{1:T}|I_0,\phi)}\left[\log \frac{p_\theta(I_{0:T})}{q(I_{1:T} \mid I_0, \phi)}\right] =: \mathcal{L}_{\text{ELBO}}. \tag{22}$$

Following the DDPM formulation, the forward transitions can be rewritten as $q(I_t \mid I_{t-1}, \phi) = q(I_t \mid I_{t-1}, I_0, \phi)$. By Bayes' rule, each transition admits the decomposition

$$q(I_t \mid I_{t-1}, I_0, \phi) = \frac{q(I_{t-1} \mid I_t, I_0, \phi)\, q(I_t \mid I_0, \phi)}{q(I_{t-1} \mid I_0, \phi)}. \tag{23}$$

Substituting into the ELBO, the objective expands as

$$-\mathcal{L}_{\text{ELBO}} = \mathbb{E}_q\left[-\log \frac{p_\theta(I_{0:T})}{q(I_{1:T} \mid I_0, \phi)}\right] \tag{24}$$

$$= \mathbb{E}_q\left[-\log p_\theta(I_T) - \sum_{t \geq 1}\log \frac{p_\theta(I_{t-1} \mid I_t)}{q(I_t \mid I_{t-1}, \phi)}\right] \tag{25}$$

$$= \mathbb{E}_q\left[-\log p_\theta(I_T) - \sum_{t > 1}\log \frac{p_\theta(I_{t-1} \mid I_t)}{q(I_t \mid I_{t-1}, \phi)} - \log \frac{p_\theta(I_0 \mid I_1)}{q(I_1 \mid I_0, \phi)}\right] \tag{26}$$

$$= \mathbb{E}_q\left[-\log p_\theta(I_T) - \sum_{t > 1}\log \frac{p_\theta(I_{t-1} \mid I_t)}{q(I_{t-1} \mid I_t, I_0, \phi)}\frac{q(I_{t-1} \mid I_0, \phi)}{q(I_t \mid I_0, \phi)} - \log \frac{p_\theta(I_0 \mid I_1)}{q(I_1 \mid I_0, \phi)}\right] \tag{27}$$

$$= \mathbb{E}_q\left[-\log \frac{p_\theta(I_T)}{q(I_T \mid I_0, \phi)} - \sum_{t > 1}\log \frac{p_\theta(I_{t-1} \mid I_t)}{q(I_{t-1} \mid I_t, I_0, \phi)} - \log p_\theta(I_0 \mid I_1)\right]. \tag{28}$$

Rewriting (28) in terms of KL divergence yields

$$-\mathcal{L}_{\text{ELBO}} \tag{29}$$

$$= \mathbb{E}_q\left[D_{\text{KL}}(q(I_T \mid I_0, \phi) \parallel p_\theta(I_T)) + \sum_{t \geq 1}D_{\text{KL}}(q(I_{t-1} \mid I_t, I_0, \phi) \parallel p_\theta(I_{t-1} \mid I_t)) - \log p_\theta(I_0 \mid I_1)\right]. \tag{30}$$

Unlike standard diffusion models that assume a standard Gaussian prior at the terminal state, our model defines the prior distribution as

$$p_\theta(I_T) = \mathcal{N}(I_T;\ I_H,\ \bar{\beta}_T^2 \mathbf{I}), \tag{31}$$

where $I_H$ is the hazy image modeled by the Atmospheric Scattering Model. In contrast to an arbitrary isotropic Gaussian prior, our prior represents a noise-perturbed hazy observation that is fully consistent with the physical forward process. Specifically, both the forward marginal distribution $q(I_T \mid I_0, \phi)$ and the prior $p_\theta(I_T)$ are Gaussian with identical mean $I_H$ and variance $\bar{\beta}_T^2$. As a result, their KL divergence is equal to zero under this design as

$$D_{\mathrm{KL}}(q(I_T \mid I_0, \phi) \,\|\, p_\theta(I_T)) = 0 \tag{32}$$

Following the standard approach in DDPM and DDIM, the training objective reduces to the sum of stepwise KL divergence terms, which quantify the discrepancy between forward and reverse transitions as

$$\sum_{t \geq 1} D_{\mathrm{KL}}(q(I_{t-1} \mid I_t, I_0, \phi) \,\|\, p_\theta(I_{t-1} \mid I_t)). \tag{33}$$

To compute these terms, we derive $q(I_{t-1} \mid I_t, I_0, \phi)$ using Bayes' rule as

$$q(I_{t-1} \mid I_t, I_0, e_t) = \frac{q(I_t \mid I_{t-1}, I_0, \phi)\, q(I_{t-1} \mid I_0, \phi)}{q(I_t \mid I_0, \phi)}. \tag{34}$$

From (13), we have

$$q(I_t \mid I_{t-1}, I_0, \phi) = \mathcal{N}\big(I_t \mid I_{t-1}\tau_t + A(1 - \tau_t),\, \beta_t^2\big), \quad \text{where} \quad \tau_t = e^{-\alpha_t \sigma d} \tag{35}$$

From (21), we have

$$q(I_t \mid I_0, \phi) = \mathcal{N}\big(I_t \mid I_0 \bar{\tau}_t + A(1 - \bar{\tau}_t),\, \bar{\beta}_t^2\big). \tag{36}$$

Combining the above, we obtain

$$q(I_{t-1} \mid I_t, I_0, e_t) = \frac{\mathcal{N}\big(I_t \mid I_{t-1}\tau_t + A(1-\tau_t), \beta_t^2\big)\mathcal{N}\big(I_{t-1} \mid I_0\bar{\tau}_{t-1} + A(1-\bar{\tau}_{t-1}), \bar{\beta}_{t-1}^2\big)}{\mathcal{N}\big(I_t \mid I_0\bar{\tau}_t + A(1-\bar{\tau}_t), \bar{\beta}_t^2\big)} \tag{37}$$

$$\propto \exp\left\{ -\left[ \frac{(I_t - (\tau_t I_{t-1} + (1-\tau_t)A))^2}{2\beta_t^2} + \frac{(I_{t-1} - (\bar{\tau}_{t-1}I_0 + (1-\bar{\tau}_{t-1})A))^2}{2\bar{\beta}_{t-1}^2} - \frac{(I_t - (\bar{\tau}_t I_0 + (1-\bar{\tau}_t)A))^2}{2\bar{\beta}_t^2} \right] \right\} \tag{38}$$

$$= \exp\left\{ -\frac{1}{2}\left[ \frac{(I_t - (\tau_t I_{t-1} + (1-\tau_t)A))^2}{\beta_t^2} + \frac{(I_{t-1} - (\bar{\tau}_{t-1}I_0 + (1-\bar{\tau}_{t-1})A))^2}{\bar{\beta}_{t-1}^2} - \frac{(I_t - (\bar{\tau}_t I_0 + (1-\bar{\tau}_t)A))^2}{\bar{\beta}_t^2} \right] \right\} \tag{39}$$

$$= \exp\left\{ -\frac{1}{2}\left[ \left(\frac{\bar{\beta}_t^2}{\beta_t^2 \bar{\beta}_{t-1}^2}\right)I_{t-1}^2 - 2\left(\frac{\tau_t I_t}{\beta_t^2} - \frac{\tau_t(1-\tau_t)A}{\beta_t^2} + \frac{\bar{\tau}_{t-1}I_0}{\bar{\beta}_{t-1}^2} + \frac{(1-\bar{\tau}_{t-1})A}{\bar{\beta}_{t-1}^2}\right)I_{t-1} + C(I_I, I_0, \phi) \right] \right\}, \tag{40}$$

where $C(I_I, I_0, \phi)$ ) denotes the terms not involving $I_{t-1}$. From (40), the mean and the variance of $q(I_{t-1} \mid I_t, I_0, e_t)$ are given by

$$\mu_t(I_t, I_0, \phi) = \frac{\frac{\tau_t I_t}{\beta_t^2} - \frac{\tau_t(1-\tau_t)A}{\beta_t^2} + \frac{\bar{\tau}_{t-1}I_0}{\bar{\beta}_{t-1}^2} + \frac{(1-\bar{\tau}_{t-1})A}{\bar{\beta}_{t-1}^2}}{\frac{\bar{\beta}_t^2}{\beta_t^2 \bar{\beta}_{t-1}^2}} \tag{41}$$

$$= \frac{I_t}{\tau_t} - \frac{1-\tau_t}{\tau_t}A - \frac{\beta_t^2}{\tau_t \bar{\beta}_t}\epsilon\,; \tag{42}$$

$$\delta_t(I_t, I_0, \phi) = \frac{\beta_t^2 \bar{\beta}_{t-1}^2}{\bar{\beta}_t^2}. \tag{43}$$

We model the reverse process beginning at

$$p_\theta(I_T) = \mathcal{N}(I_T;\ I_H,\ \bar{\beta}_T^2 \mathbf{I}), \tag{44}$$

and define

$$p_\theta(I_{t-1} \mid I_t) = q(I_{t-1} \mid I_t, I_0, e_t). \tag{45}$$

In our setting, since the variances of the two Gaussian distributions are matched exactly, the KL divergence reduces to a squared difference between their means, as is standard in DDPM. Accordingly, the KL divergence term in (33) reduces to

$$D_{\mathrm{KL}}(q(I_{t-1} \mid I_t, I_0, \phi) \parallel p_\theta(I_{t-1} \mid I_t)) = \mathbb{E}\left[\left\| \mu_t - \mu_t^\theta \right\|^2\right], \tag{46}$$

where the mean of the true posterior is given by

$$\mu_t = \frac{I_t}{\tau_t} - \frac{1-\tau_t}{\tau_t} A - \frac{\beta_t^2}{\tau_t \bar{\beta}_t} \epsilon, \tag{47}$$

and the model-predicted mean is

$$\mu_t^\theta = \frac{I_t}{\tau_t^\theta} - \frac{1-\tau_t^\theta}{\tau_t^\theta} A^\theta - \frac{\beta_t^2}{\tau_t^\theta \bar{\beta}_t} \epsilon^\theta, \tag{48}$$

with learnable estimators $\tau_t^\theta$, $A^\theta$, and $\epsilon^\theta$.

Previous diffusion-based approaches (Chen et al., 2023; Xia et al., 2023; Salimans & Ho, 2022; Rao et al., 2024) typically supervise only the terminal reconstruction and backpropagate through the entire reverse chain, thereby amortizing step-wise supervision. Following this paradigm and leveraging (46), we parameterize the reverse transition mean with three estimators $\tau_t^\theta$, $A^\theta$, and $\epsilon^\theta$. During training, we sample $I_T \sim p_\theta(I_T)$ and iteratively apply the learned reverse transitions $p_\theta(I_{t-1} \mid I_t)$—whose mean is $\mu_t^\theta(I_t; \tau_t^\theta, A^\theta, \epsilon^\theta)$—to denoise step by step from $I_T \to I_{T-1} \to \cdots \to I_0^\theta$. We then supervise only the final output, explicitly minimizing the reconstruction discrepancy

$$\mathcal{L}_{\mathrm{final}} = \| I_0 - I_0^\theta \|, \tag{49}$$

so that gradients propagate through the full reverse trajectory and drive $\{\tau_t^\theta, A^\theta, \epsilon^\theta\}$ to produce consistent denoising updates toward the clean image.

However, this formulation does not explicitly incorporate haze characteristics into the noise scheduling, and the reliance on three separate estimators introduces additional training overhead. To address these limitations, we introduce a *haze-aware noise scheduler* that dynamically adjusts $\beta_t$ according to haze density, thereby achieving more effective and physically grounded noise scheduling, as described in Section 3.1. Furthermore, when combined with our derived deterministic implicit sampling formulation, this approach enables us to remove the dependency on $A^\theta$ and reduce the reverse parameterization to only two estimators, $\tau_t^\theta$ and $\epsilon^\theta$, significantly simplifying the learning process. The detailed description is provided in Appendix A.3.

### A.3 DETERMINISTIC IMPLICIT SAMPLING FOR HNDIFF

In this section, following the induction-based argument of DDIM (Song et al., 2021), we derive a deterministic reverse process for HNDiff and prove that it preserves the forward marginal distribution defined in (21). Specifically, recall that the forward marginal is given by

$$q(I_t \mid I_0, \phi) = \mathcal{N}\big(I_t; \bar{\tau}_t I_0 + (1-\bar{\tau}_t)A, \ \bar{\beta}_t^2 \mathbf{I}\big), \tag{50}$$

where $\bar{\tau}_t = \prod_{s=1}^t \tau_s$ and $\bar{\beta}_t^2 = \sum_{s=1}^t \left(\prod_{j=s+1}^t \tau_j^2\right)\beta_s^2$.

Following the deterministic implicit sampling formulation of DDIM, we define the reverse transition distribution as

$$q_\delta(I_{t-1} \mid I_t, I_0, \phi) = \mathcal{N}\big(I_{t-1}; \mu_{t-1}, \ \delta_t^2 \mathbf{I}\big), \tag{51}$$

with mean

$$\mu_{t-1} = \bar{\tau}_{t-1} I_0 + (1-\bar{\tau}_{t-1})A + \sqrt{\bar{\beta}_{t-1}^2 - \delta_t^2}\ \frac{I_t - (\bar{\tau}_t I_0 + (1-\bar{\tau}_t)A)}{\bar{\beta}_t}, \tag{52}$$

and variance

$$\delta_t^2 = \eta \cdot \frac{\beta_t^2 \bar{\beta}_{t-1}^2}{\bar{\beta}_t^2}, \tag{53}$$

where $\eta \in [0, 1]$ controls the sampling stochasticity. Setting $\eta = 0$ yields a purely deterministic sampler.

**Induction proof of consistency.** We now prove by induction, as in DDIM, that the above reverse process preserves the forward marginal distribution. For the base case $t = T$, the marginal $q(I_T \mid I_0, \phi)$ is valid by definition. Assume that $q(I_t \mid I_0, \phi)$ holds at step $t$. Then, sampling $I_{t-1}$ from $q_\delta(I_{t-1} \mid I_t, I_0, \phi)$ yields mean

$$\mathbb{E}[I_{t-1} \mid I_0, \phi] = \bar{\tau}_{t-1}I_0 + (1 - \bar{\tau}_{t-1})A, \tag{54}$$

and variance

$$\delta_{t-1}^2 \mathbf{I} = \delta_t^2 \mathbf{I} + \left( \frac{\sqrt{\bar{\beta}_{t-1}^2 - \delta_t^2}}{\bar{\beta}_t} \right)^2 \bar{\beta}_t^2 \mathbf{I} \tag{55}$$

$$= \bar{\beta}_{t-1}^2 \mathbf{I}. \tag{56}$$

Thus,

$$q(I_{t-1} \mid I_0, \phi) = \mathcal{N}\left( I_{t-1};\ \bar{\tau}_{t-1}I_0 + (1 - \bar{\tau}_{t-1})A,\ \bar{\beta}_{t-1}^2 \mathbf{I} \right), \tag{57}$$

which confirms that the forward distribution holds at step $t-1$. By induction, the deterministic sampler remains consistent with the forward process across all time steps.

**Deterministic Implicit Sampling.** For deriving the deterministic implicit sampling formulation from $I_t$ to $I_{t-1}$, we first define $I_{t-1}$ from (51) as

$$I_{t-1} = \bar{\tau}_{t-1}I_0 + (1 - \bar{\tau}_{t-1})A + \sqrt{\bar{\beta}_{t-1}^2 - \delta_t^2} \cdot \frac{I_t - (\bar{\tau}_t I_0 + (1 - \bar{\tau}_t)A)}{\bar{\beta}_t} + \delta_t, \tag{58}$$

where $\delta_t$ is a variance parameter controlling sampling stochasticity, and $\delta_t = 0$ corresponds to deterministic sampling. Next, by expressing $I_0$ as

$$I_0 = \frac{I_t - (1 - \bar{\tau}_t)A - \bar{\beta}_t \epsilon_t}{\bar{\tau}_t}, \tag{59}$$

based on (21) and substituting it into (58), we obtain the simplified deterministic update rule:

$$I_{t-1} = \frac{I_t - A(1 - \tau_t) - \beta_t \epsilon_t}{\tau_t}. \tag{60}$$

**Reduction of Reverse Parameterization.** As discussed in Section 3.1, we adopt a haze-aware noise scheduler by setting $\beta_t = 1 - \tau_t$, such that the injected noise is explicitly modulated by the haze density. Substituting this into (60), the reverse update becomes

$$I_{t-1} = \frac{I_t - A(1 - \tau_t) - \epsilon_t(1 - \tau_t)}{\tau_t} = \frac{I_t - (A + \epsilon_t)(1 - \tau_t)}{\tau_t} = \frac{I_t - N_t(1 - \tau_t)}{\tau_t}, \tag{61}$$

where $N_t = A + \epsilon_t$ denotes the *atmospheric noise*, which combines the atmospheric light term and a Gaussian perturbation.

Under this formulation, the reverse dynamics require only *two learnable estimators*: (1) a noise estimator $N_t^\theta(I_t, I_H, t)$ that approximates the atmospheric noise $N_t$, and (2) a haze estimator $1 - \tau_t^\theta(I_t, I_H, t)$ that directly models the haze residual $1 - \tau_t$.

This reduction naturally emerges from combining the deterministic implicit sampling rule with the haze-aware noise scheduler, thereby simplifying the parameterization of the reverse process while preserving consistency with the forward distribution and avoiding the need for an explicit $A^\theta$ estimator.

## A.4 Dehazing results on RTTS dataset

We have conducted additional experiments on the RTTS dataset. Our models are pretrained on the NH-HAZE dataset and directly evaluated on RTTS to assess out-of-domain generalization. As shown in Table 8, the HNDiff-enhanced models consistently achieve lower BRISQUE and NIQE scores than their corresponding baselines, indicating better perceptual quality on this real-world benchmark. Moreover, we further apply HNDiff to the ASM-based dehazing method RIDCP (Wu et al., 2023) to verify its compatibility with existing physical-model approaches. The dehazing results are reported in Table 9, the corresponding object-detection mAP on RTTS is summarized in Table 10, and qualitative comparisons are shown in Figure 22, all of which demonstrate that HNDiff can consistently enhance RIDCP in both restoration quality and downstream perception.

Table 8: Results on RTTS, where the proposed HNDiff is applied to three baselines: FocalNet, ConvIR, and SGDN.

| Metric | FocalNet | HNDiff (FocalNet) | ConvIR | HNDiff (ConvIR) | SGDN | HNDiff (SGDN) |
|---|---|---|---|---|---|---|
| BRISQUE ↓ | 35.9789 | **29.8278** | 36.5144 | **34.9876** | 34.6808 | **32.5563** |
| NIQE ↓ | 4.3392 | **4.2867** | 4.3992 | **4.2665** | 4.8426 | **4.6343** |

Table 9: Quantitative comparison of RIDCP and HNDiff (RIDCP) on RTTS.

| Method | FADE↓ | NIMA↑ | BRISQUE↓ |
|---|---|---|---|
| RIDCP | 0.944 | 4.97 | 17.29 |
| HNDiff (RIDCP) | **0.417** | **5.08** | **16.09** |

## A.5 DEHAZING RESULTS OF DIFFERENT DIFFUSION MECHANISMS.

We present additional dehazed results of different diffusion mechanisms, including DDPM (Ho et al., 2020), RDDM (Liu et al., 2024a), and our proposed HNDiff, on the NH-HAZE test set. As shown in Figures 7 and 8, we further facilitate visual comparison by computing residual maps between the outputs and the ground truth, where each residual map is normalized by the global maximum residual value across all methods to ensure consistent scaling. Brighter regions indicate larger discrepancies from the ground truth. Compared to DDPM and RDDM, HNDiff produces cleaner reconstructions with notably lower residual intensities, demonstrating its superior dehazing capability.

## A.6 DEHAZING RESULTS ON REAL-WORLD DATASETS

We present additional dehazed results on four real-world datasets to compare models integrated with HNDiff against their original counterparts. Qualitative evaluations are conducted on three representative image dehazing networks, namely FocalNet, ConvIR, and SGDN. For the NH-HAZE dataset, the comparisons are shown in Figures 9, 10, 11, and 12. For the $RW^2AH$ dataset, the comparisons are shown in Figures 13, 14, and 15. For the Dense-HAZE dataset, the comparisons are shown in Figure 16 and Figure 17. For the O-HAZE dataset, we further enhance the visual comparison by computing residual maps between the baseline outputs and the ground truth, as well as between the HNDiff outputs and the ground truth. Both residual maps are normalized using the global maximum residual value across the two maps to ensure consistent scaling. In this visualization, brighter regions indicate larger discrepancies from the ground truth, as illustrated in Figures 18, 19, and 20.

## A.7 COMPARISON OF DEHAZING RESULTS WITH PRIOR METHODS.

We further provide qualitative comparisons on the real-world hazy dataset $RW^2AH$. Figure 21 shows visual results of previous dehazing methods and our HNDiff. Compared to prior approaches, our method produces clearer structures, more natural colors, and fewer artifacts on challenging real-world hazy images, demonstrating a noticeable qualitative improvement and validating the effectiveness of HNDiff in practical scenarios.

## A.8 ARCHITECTURE OF THE HAZE ESTIMATOR U-NET

Our haze estimator adopts a time-conditional U-Net with four resolution scales. The network takes two 4-channel latent features as input, concatenates them into an 8-channel tensor, and feeds them to a $7 \times 7$ convolution (padding 3) to obtain a 32-channel feature map. We use a base width of 32 and a channel progression of $(32, 64, 128, 128)$ across scales. At each encoder stage, we apply two time-conditioned residual blocks (ResNet blocks with group normalization and nonlinearity) followed by a linear-attention block in a residual form, and then downsample the feature map (strided convolution, except at the last stage where a $3 \times 3$ convolution keeps the resolution). The diffusion timestep is embedded by a sinusoidal positional embedding followed by a two-layer MLP of width $4 \dim$, and this time embedding is injected into all residual blocks in both the encoder and decoder. At the bottleneck, we use a ResNet block, a full self-attention block, and another ResNet block at 128 channels. The decoder mirrors the encoder: at each scale, we concatenate the current feature with the

Table 10: Object detection performance (mAP@50) on RTTS using a pretrained YOLOv3.

| Method | person | bicycle | car | motorcycle | bus | mean |
|---|---|---|---|---|---|---|
| Hazy Image | 0.662 | 0.425 | 0.581 | 0.376 | 0.299 | 0.469 |
| RIDCP | 0.669 | 0.444 | 0.611 | 0.448 | 0.341 | 0.503 |
| HNDiff (RIDCP) | **0.677** | **0.454** | **0.629** | **0.452** | **0.361** | **0.515** |

corresponding encoder feature (standard U-shaped skip connections), apply two time-conditioned residual blocks and a linear-attention block, and then upsample (except at the final stage, which uses a $3 \times 3$ convolution). Finally, the decoded feature is concatenated with the early feature from the initial $7 \times 7$ convolution, passed through one last time-conditioned residual block, and projected by a $1 \times 1$ convolution to produce the output haze/noise estimation map.

## A.9 LLM USAGE

We used a large language model (LLM) only to polish grammar and improve readability. All research ideas, methods, and results are solely from the authors.

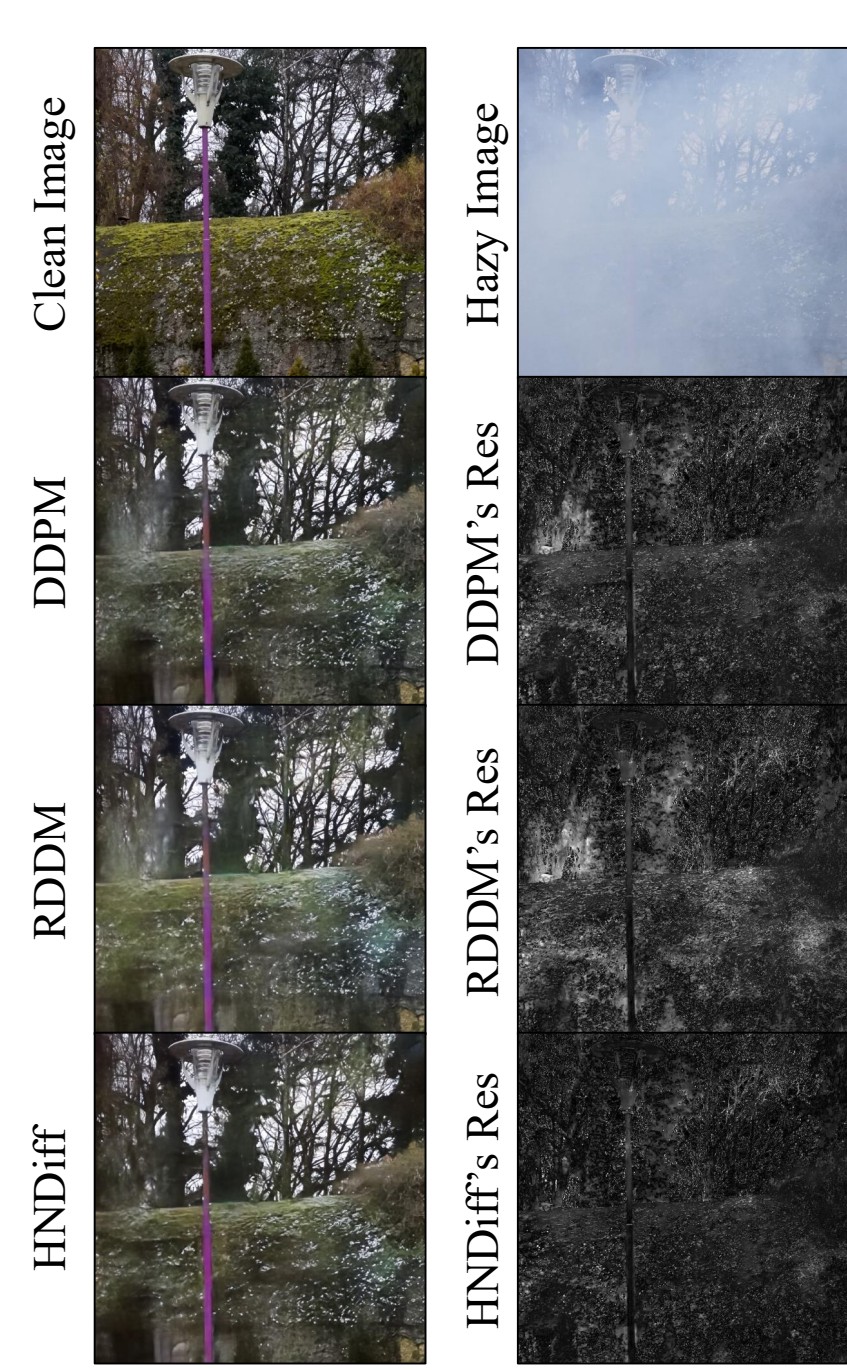

Figure 7: Dehazed results of each diffusion mechanism on NH-HAZE datasets. "Res" denotes residual maps between outputs and ground truth, where darker intensities indicate smaller errors.

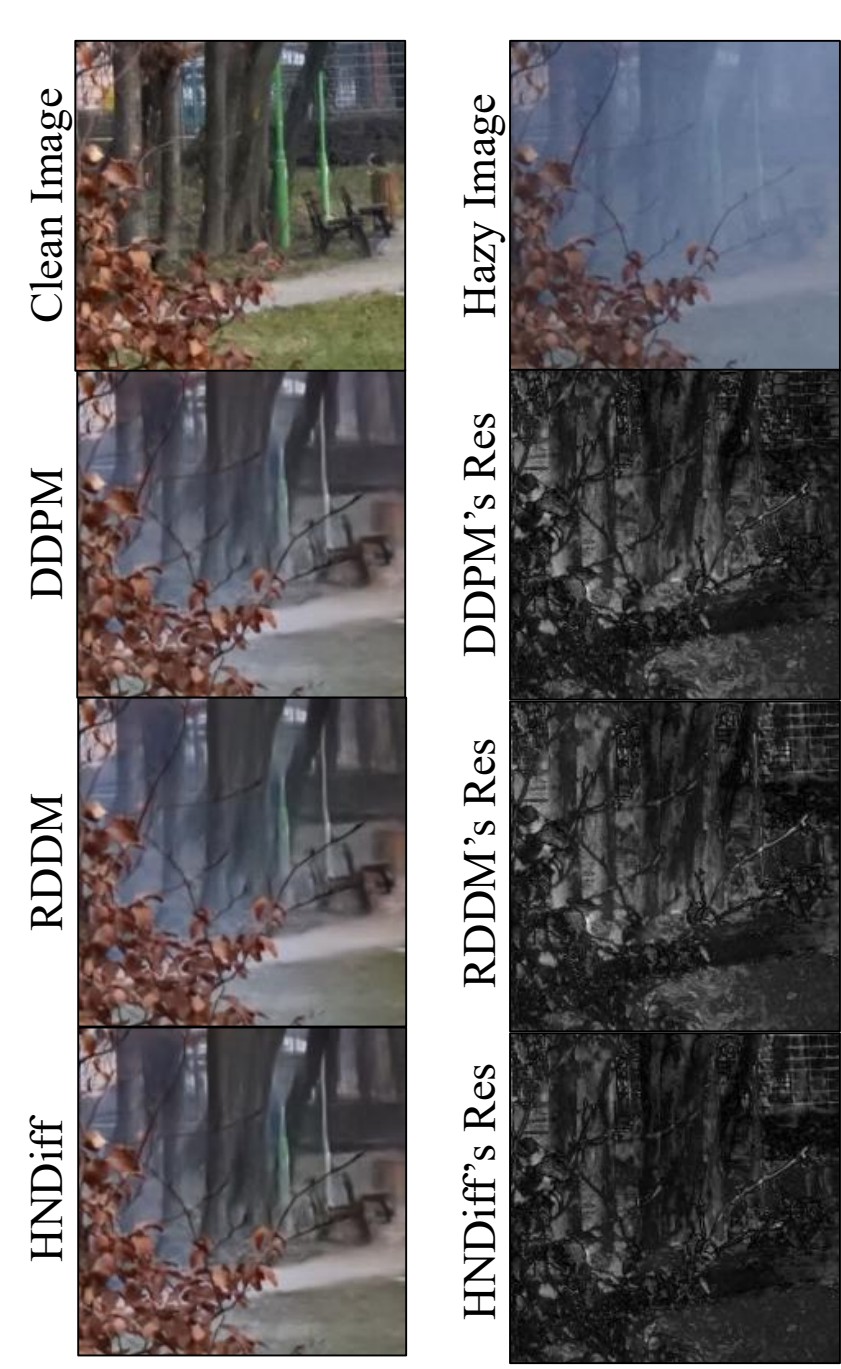

Figure 8: Dehazed results of each diffusion mechanism on NH-HAZE datasets. "Res" denotes residual maps between outputs and ground truth, where darker intensities indicate smaller errors.

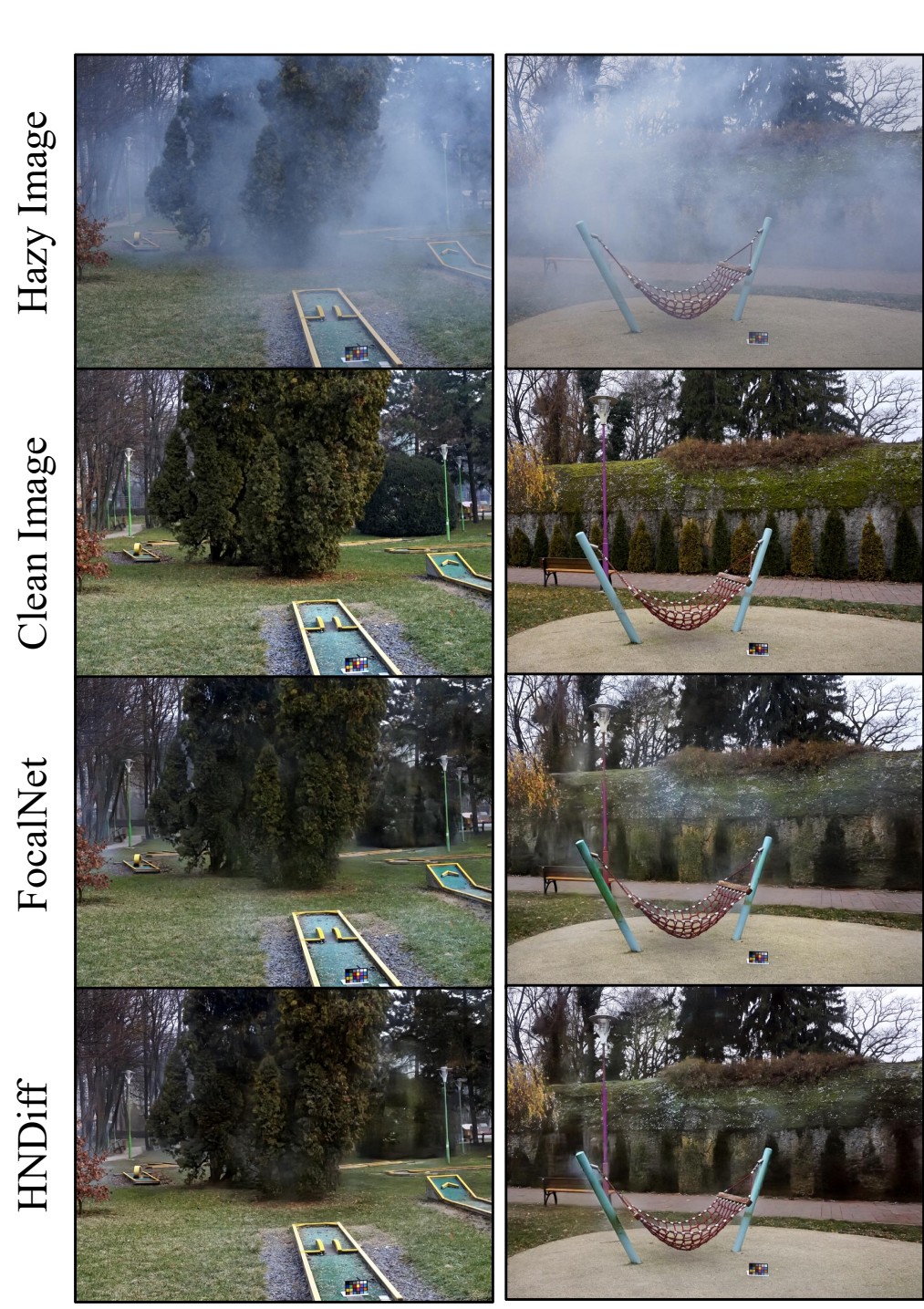

Figure 9: Qualitative results on the NH-HAZE test set.

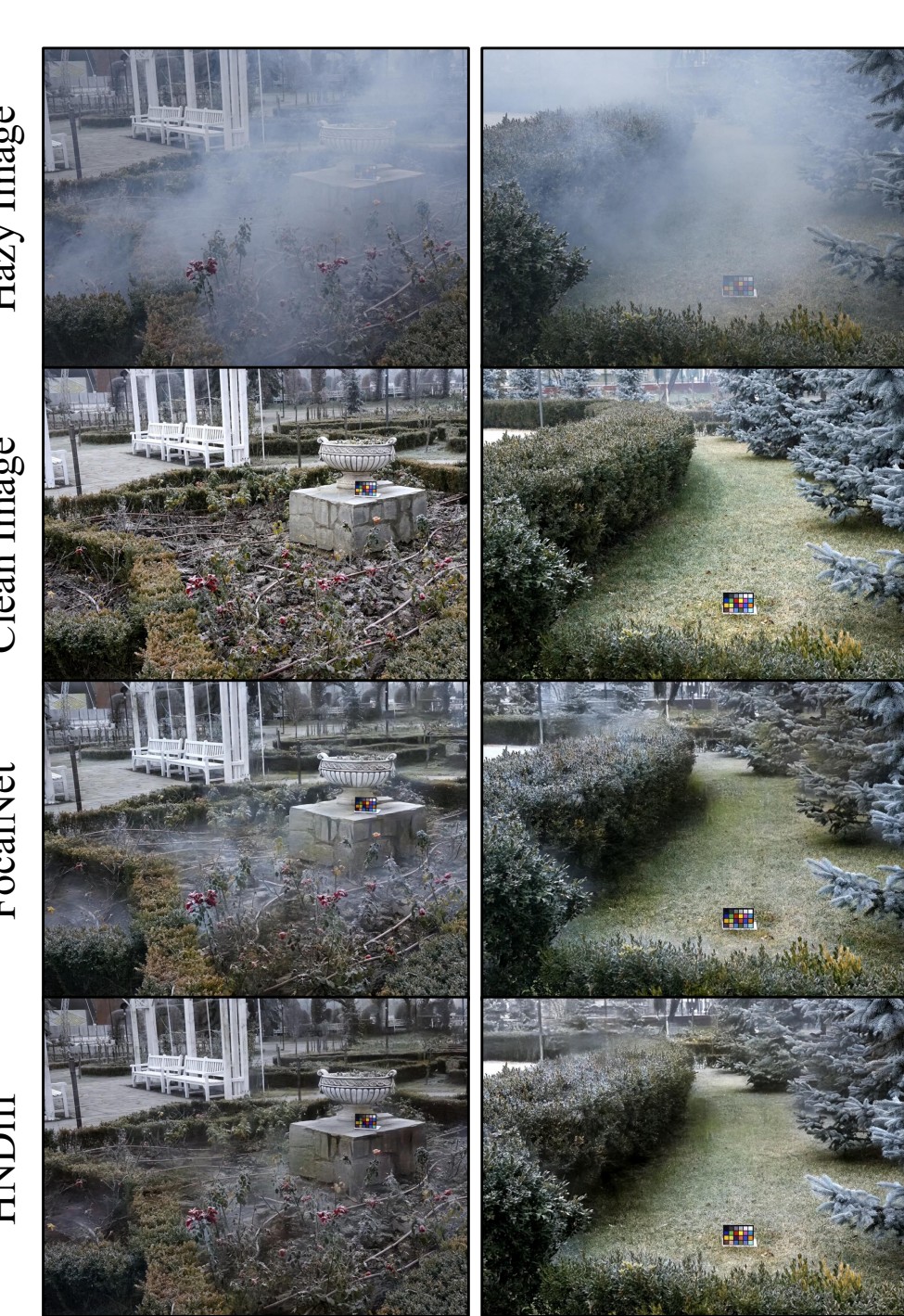

Figure 10: Qualitative results on the NH-HAZE test set.

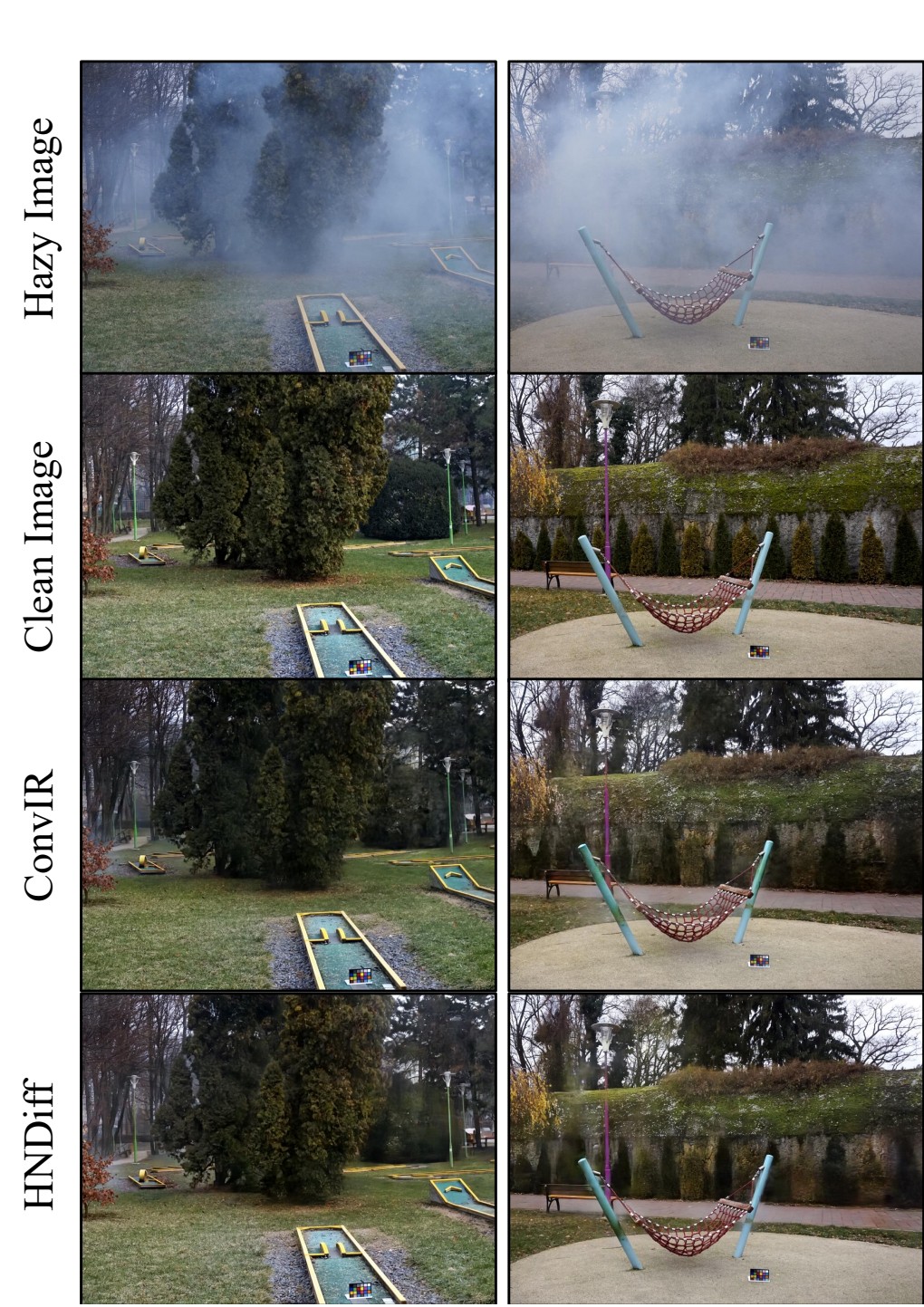

Figure 11: Qualitative results on the NH-HAZE test set.

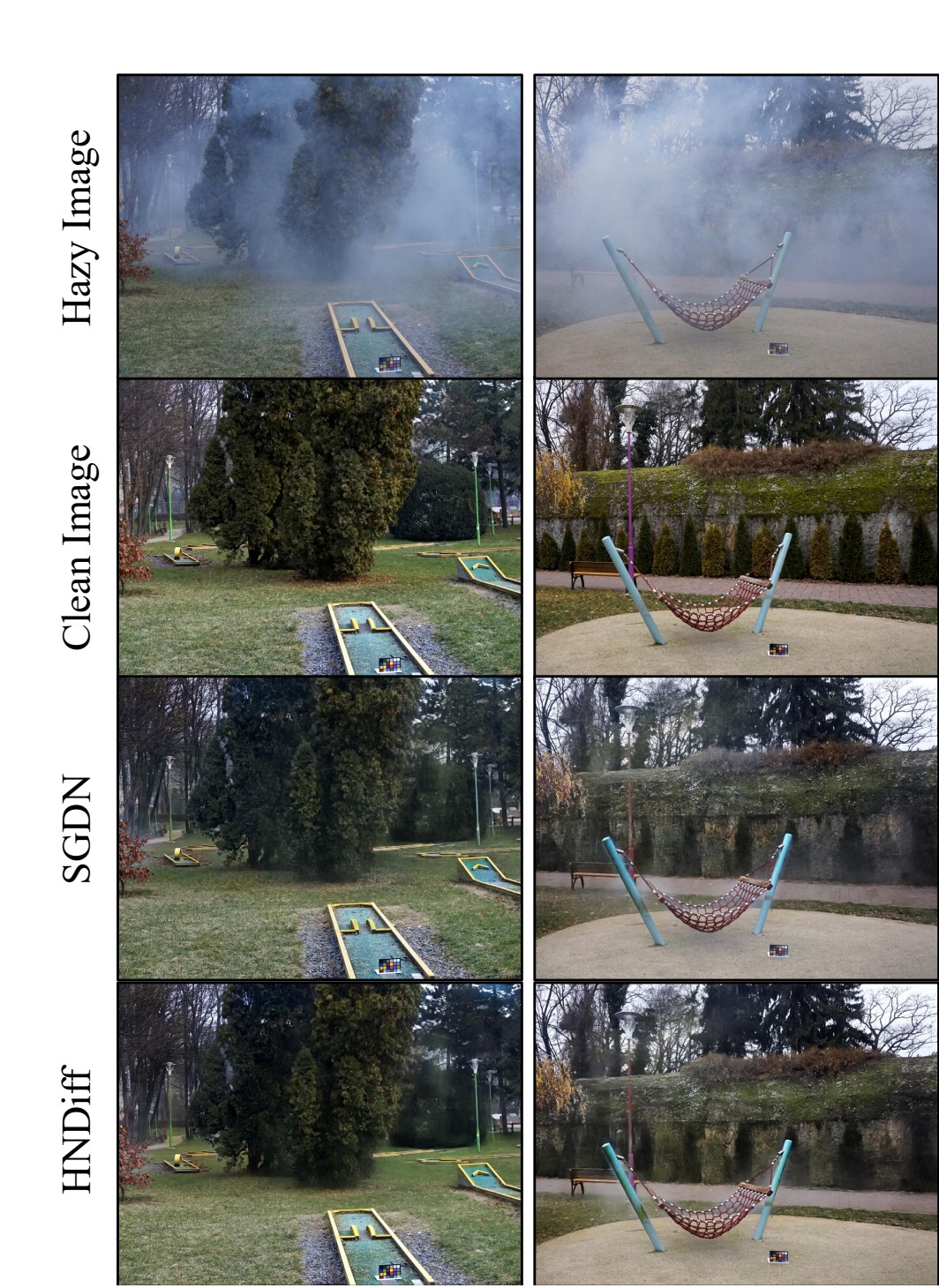

Figure 12: Qualitative results on the NH-HAZE test set.

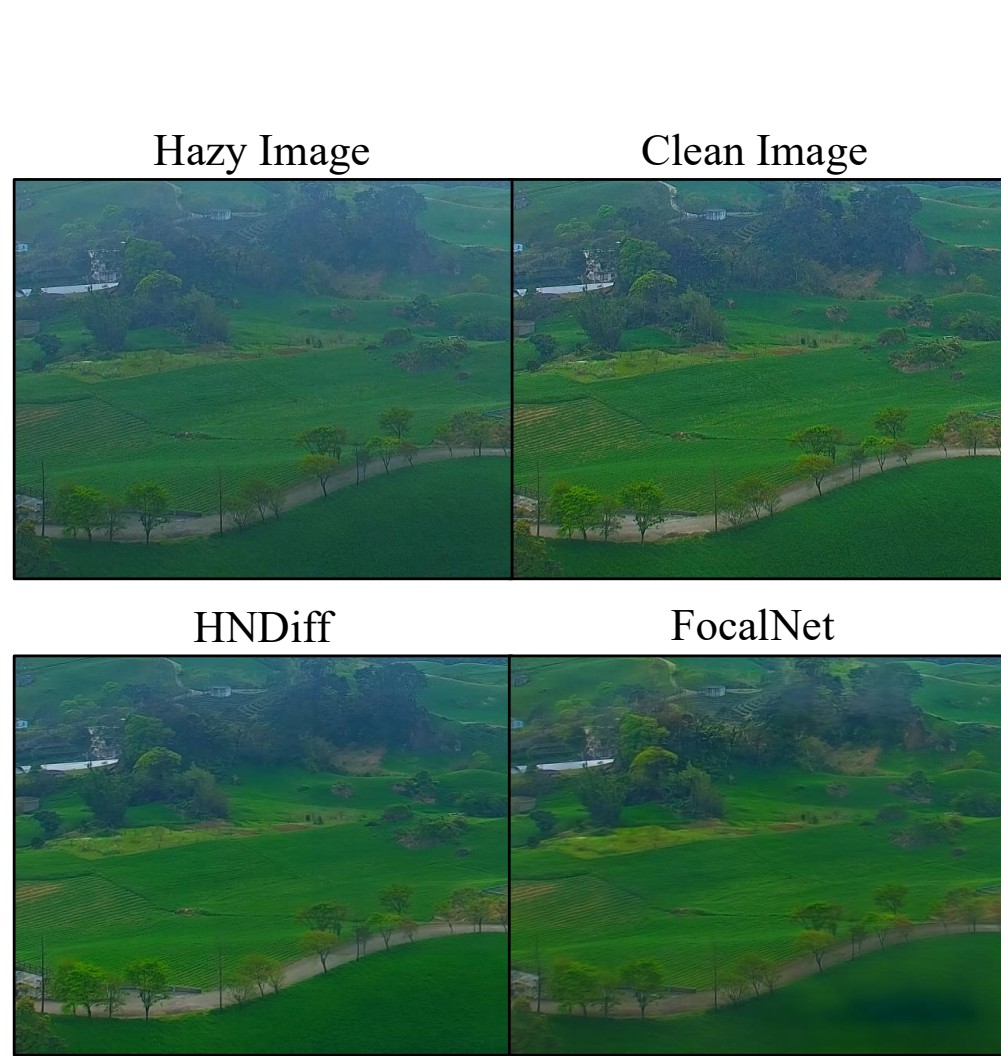

Figure 13: Qualitative results on the RW$^2$AH test set.

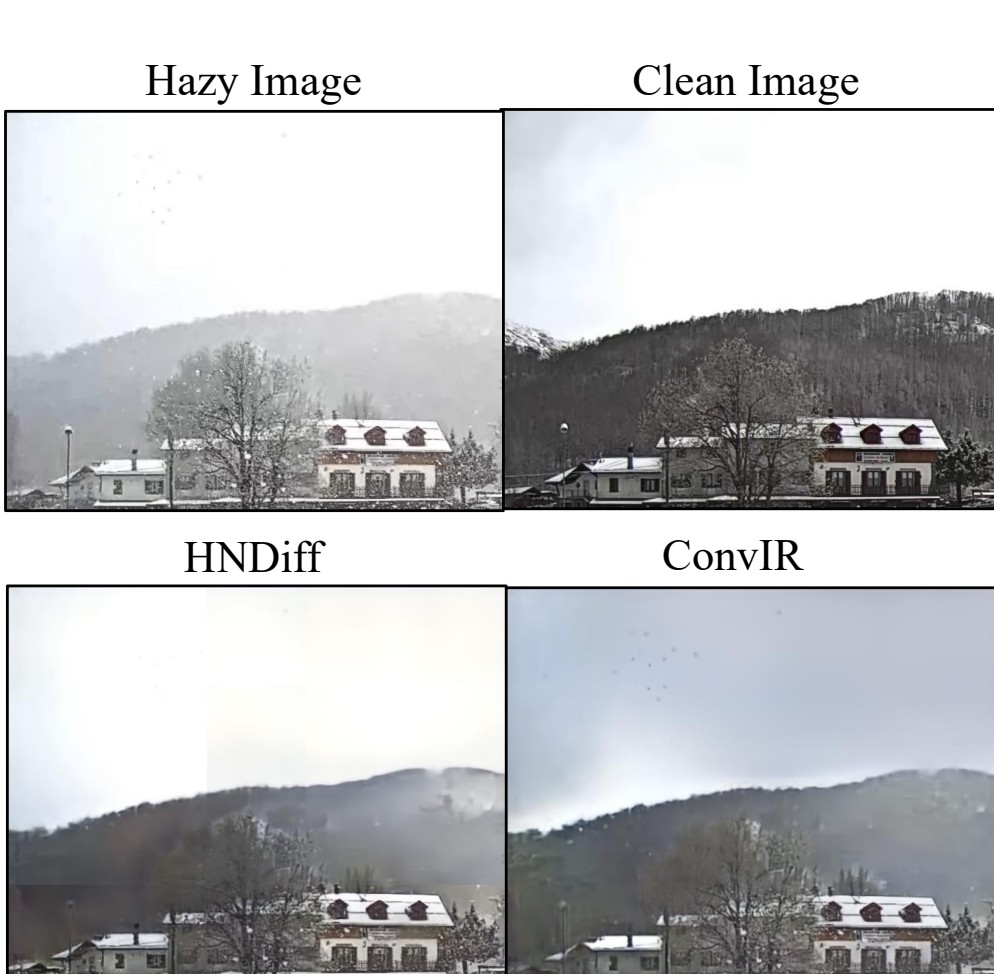

Figure 14: Qualitative results on the RW$^2$AH test set.

Hazy Image | Clean Image

HNDiff | SGDN

Figure 15: Qualitative results on the RW$^2$AH test set.

Hazy Image      Clean Image

HNDiff      FocalNet

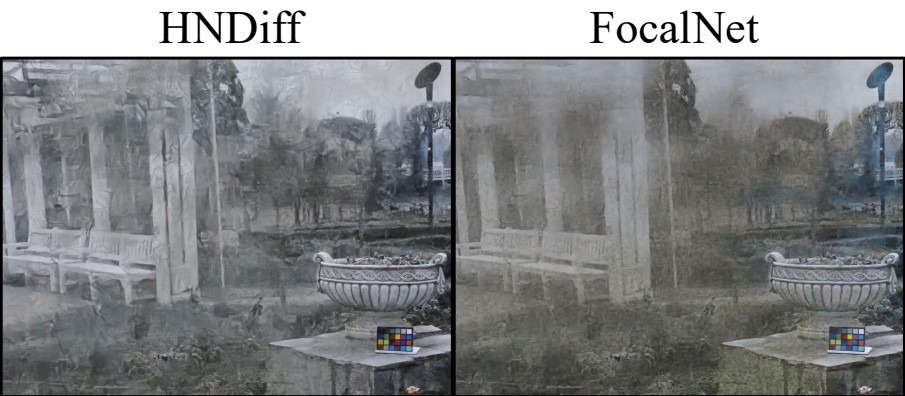

Figure 16: Qualitative results on the Dense-HAZE test set.

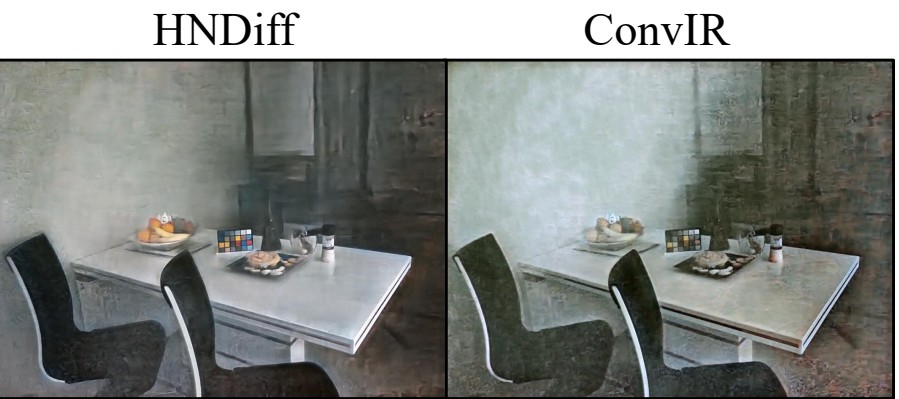

Figure 17: Qualitative results on the Dense-HAZE test set.

Hazy Image          Clean Image

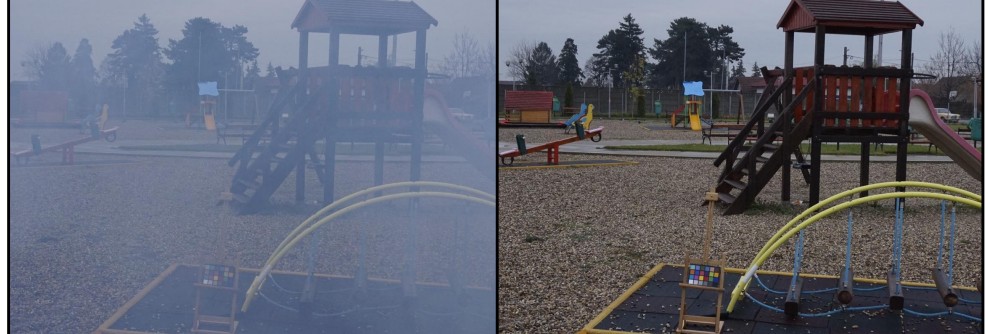

HNDiff & Residual          FocalNet & Residual

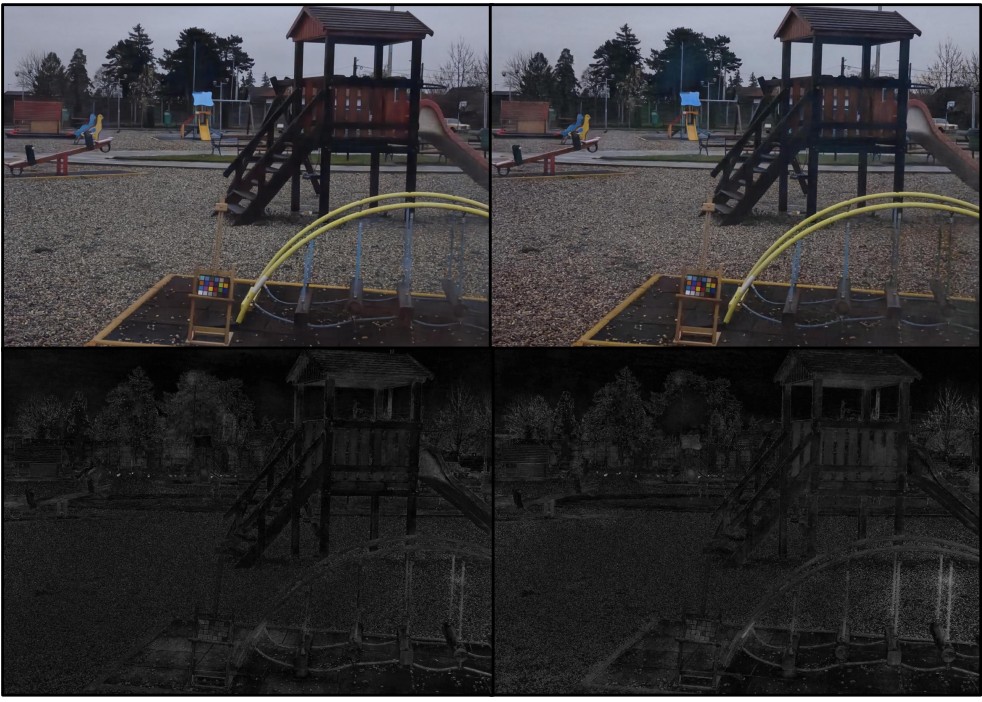

Figure 18: Qualitative results on the O-HAZE test set.

Hazy Image | Clean Image

HNDiff & Residual | ConvIR & Residual

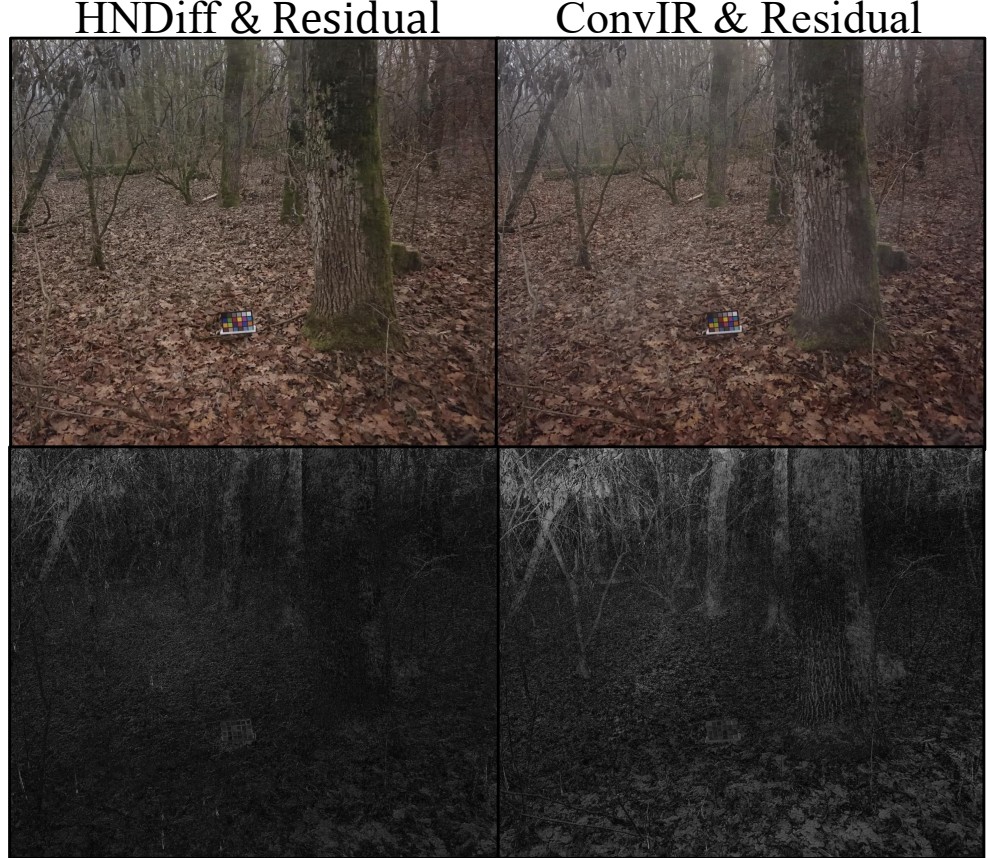

Figure 19: Qualitative results on the O-HAZE test set.

Hazy Image    Clean Image

HNDiff & Residual  SGDN & Residual

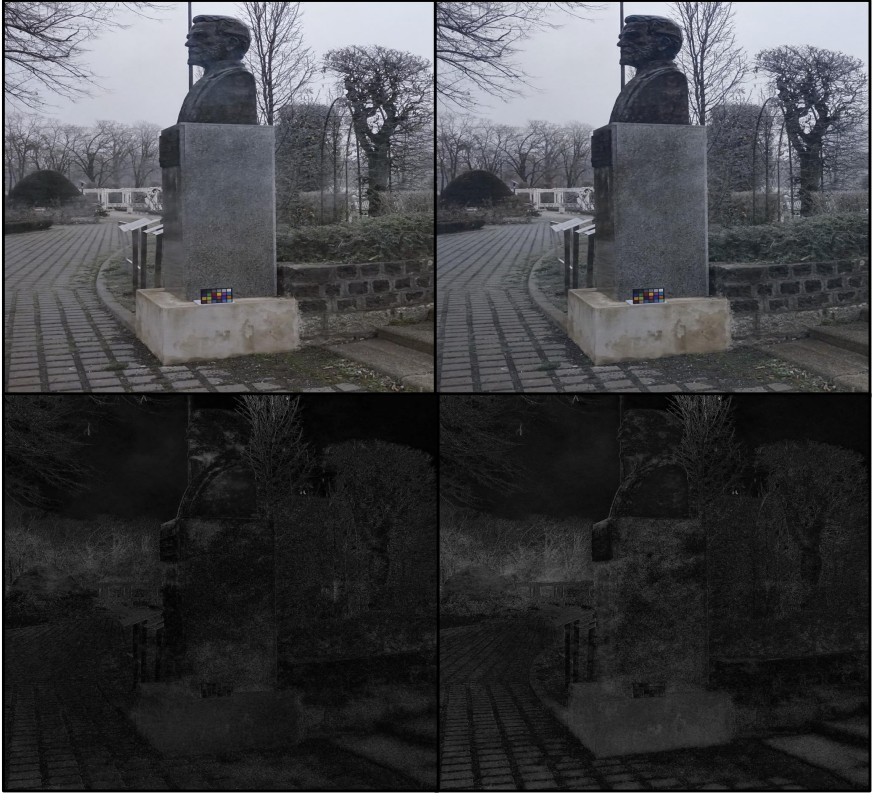

Figure 20: Qualitative results on the O-HAZE test set.

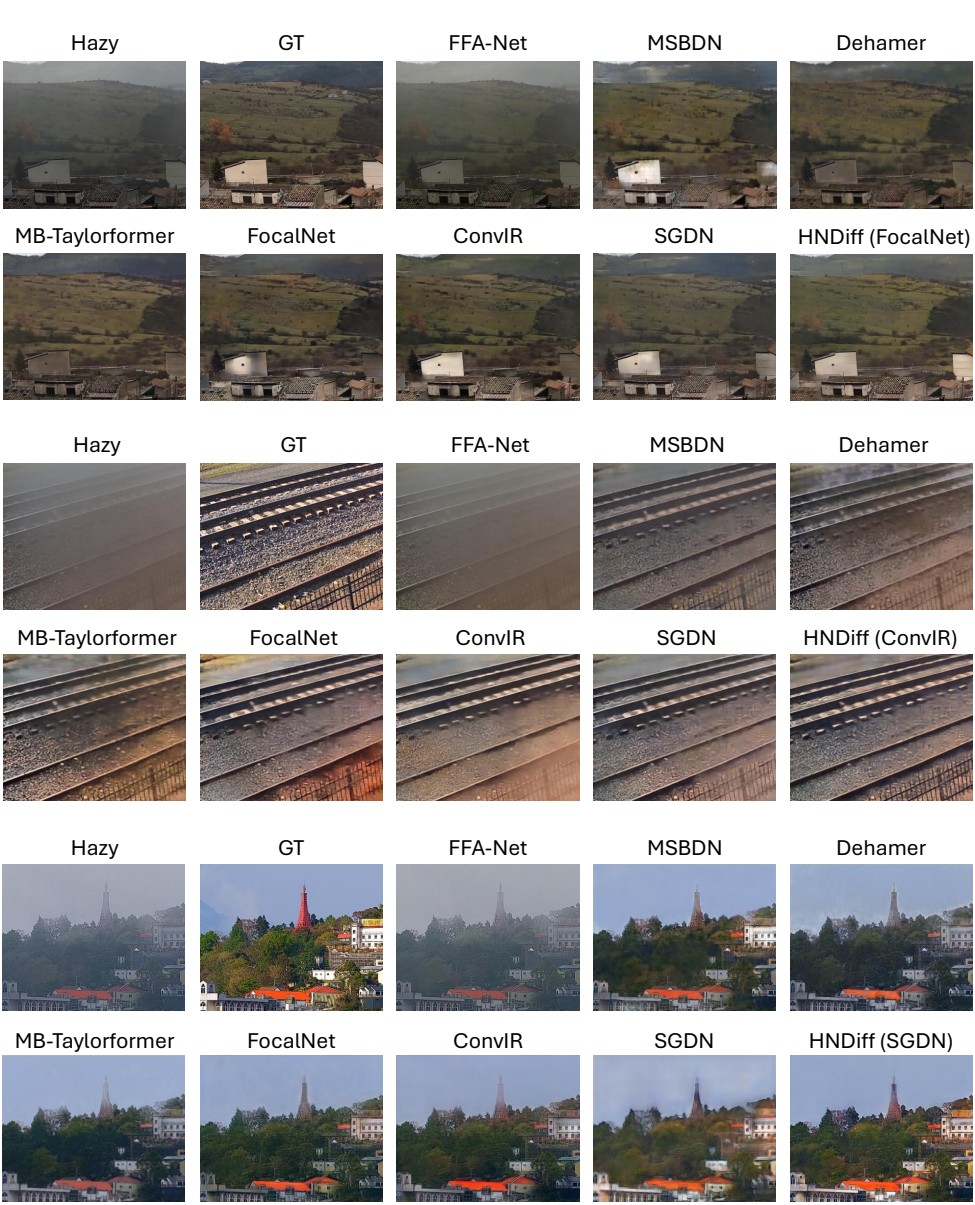

Figure 21: Comparison of Dehazing Results with prior methods.

| Hazy Image | RIDCP | HNDiff (RIDCP) |
|---|---|---|
| FADE: 1.05
NIMA: 4.92
BRISQUE: 13.29 | FADE: 0.19
NIMA: 5.13
BRISQUE: 41.02 | **FADE: 0.16**
**NIMA: 5.27**
**BRISQUE: 28.77** |
| FADE: 3.00
NIMA: 5.32
BRISQUE: 42.74 | FADE: 0.59
NIMA: 5.61
BRISQUE: 26.33 | **FADE: 0.46**
**NIMA: 5.80**
**BRISQUE: 22.51** |
| FADE: 1.41
NIMA: 5.07
BRISQUE: 7.88 | FADE: 0.52
NIMA: 5.62
BRISQUE: 6.37 | **FADE: 0.34**
**NIMA: 5.66**
**BRISQUE: 5.59** |

Figure 22: Comparison of Dehazing Results with RIDCP on RTTS dataset.

