# OpenReview forum: "HNDiff: Haze-Noise Diffusion for Image Dehazing"
_ICLR.cc/2026/Conference — Submitted to ICLR 2026_

### Official Review · Reviewer_HBqs · 2025-10-25

**Soundness:** 3
**Presentation:** 2
**Contribution:** 2
**Rating:** 4
**Confidence:** 5

**Summary:**

The paper introduces HNDiff, a diffusion-based image dehazing framework that embeds the ASM into both the diffusion processes. Instead of reconstructing clean images from pure Gaussian noise, HNDiff models haze formation by progressively adding both haze and noise in a physically grounded forward process and removes them in the reverse process. The method features a haze-aware noise scheduler that adapts noise injection based on haze density. To improve computational efficiency and generalization, a Latent HNDiff variant is proposed, operating in latent space as a prior generator integrated into standard dehazing networks through a FGM. Extensive experiments across five benchmarks and three baselines show consistent improvements in fidelity metrics.

**Strengths:**

1. Integrating the ASM as an inductive bias provides a theoretically consistent link between physical haze formation and diffusion modeling.

2. The haze-aware noise scheduler adapts the diffusion process spatially, aligning generative noise injection with haze density and improving detail recovery.

3. Comprehensive experiments on multiple datasets and architectures demonstrate consistent performance gains and strong generalization.

**Weaknesses:**

1. The three-stage training (ground-truth prior pretraining, latent diffusion optimization, joint fine-tuning) adds significant implementation difficulty and training time. The quantitative gains (e.g., +0.5 dB PSNR on average) are relatively small considering the model’s complexity and multi-stage training strategy.

2. The paper provides strong empirical results but lacks ablation or convergence analysis explaining how ASM integration affects the diffusion dynamics mathematically.

3. A method that uses a latent diffusion to provide prior to help the image restoration performance has become relatively common. This work appears to be just an implementation and improvement of this type of pattern under an atmospheric scattering model. The diffusion  process designed for mist-noise and ASM originally would have brought better interpretability, but using the potential diffusion model seems to obscure this advantage.

**Questions:**

1. How does the latent diffusion representation compare to full-image diffusion in terms of interpretability and potential loss of fine spatial information?

2. Is the improvement mainly due to the physical prior or the additional parameters introduced by the latent modules and FGM?

3. I am more curious about whether the haze-noise diffusion pattern designed by this paper can be applied to image-to-image diffusion models rather than latent diffusion models. This seems to be an interesting exploration.

4. More experiments and comparison on RTTS dataset is recommended to validate the out-of-domain generalization performance.

---

> ### Author Response · Authors · 2025-11-20
> **Rebuttal by Authors (Weakness 1 to 2)**
>
> We thank the reviewer for recognizing the theoretical design of our approach and its significant performance gains shown in experiments. We greatly appreciate these encouraging comments. We address each concern in detail below and have revised the paper accordingly.
>
> ### **Weakness 1-1: Implementation difficulty of three-stage training strategy.**
>
> The three-stage training scheme (ground-truth prior pretraining, latent diffusion optimization, and joint fine-tuning) follows the standard practice in recent latent diffusion-based image restoration methods [A, B, C], where it has been shown to be effective and stable in practice. To ensure reproducibility and minimize the implementation burden, we will release the full source code, including the complete three-stage training pipeline, upon the paper's acceptance.
>
> **Reference**
> #### [A] Chen Rao, Guangyuan Li, Zehua Lan, Jiakai Sun, Junsheng Luan, Wei Xing, Lei Zhao, Huaizhong Lin, Jianfeng Dong, and Dalong Zhang. Rethinking video deblurring with wavelet-aware dynamic transformer and diffusion model. In ECCV, 2024.
>
> #### [B] Zheng Chen, Yulun Zhang, Ding Liu, Jinjin Gu, Linghe Kong, Xin Yuan, et al. Hierarchical integration diffusion model for realistic image deblurring. In NeurIPS, 2023.
>
> #### [C] Bin Xia, Yulun Zhang, Shiyin Wang, Yitong Wang, Xinglong Wu, Yapeng Tian, Wenming Yang, and Luc Van Gool. Diffir: Efficient diffusion model for image restoration. In ICCV, 2023.
>
> ### **Weakness 1-2: The quantitative gains (e.g., +0.5 dB PSNR on average) are relatively small considering the model’s complexity and multi-stage training strategy.**
>
> In the following, we address this concern from two perspectives: the effectiveness of the three-stage training strategy and the annual performance gain of recent works.
>
> **Effectiveness of three-stage training.**
> To isolate the effect of the latent module and the three-stage training scheme, Table 4 in the paper compares different prior generators under the same configuration, including the latent module, FGM, three-stage training, and dehazing backbone. In this controlled setup, alternative priors bring at most a $0.1$ dB PSNR improvement, whereas our HNDiff prior yields a gain of $0.53$ dB. This suggests that the primary performance improvement is attributable to the proposed haze–noise diffusion design rather than merely the training protocol or latent module itself.
>
> **Annual performance gain of recent works.**
> We further clarify that an average gain of around $0.5$ dB PSNR is substantial in the current dehazing regime. As evidenced in Table 1 of the paper, recent state-of-the-art methods over the past three years improve performance on NH-HAZE by only about $0.3$ dB (FocalNet vs. ConvIR), and on O-HAZE by about $0.41$ dB (MB-TaylorFormer vs. FocalNet). In this context, the consistent ~0.5 dB improvement brought by HNDiff across multiple backbones and datasets represents a substantial advance rather than a marginal gain.
>
> ### **Weakness 2: Convergence analysis for the ASM-integrated diffusion.**
>
> Our convergence analysis for the ASM-integrated diffusion is thoroughly provided in Appendix A.2 (from Eq. (22) to Eq. (49)). Appendix A.2 starts from the standard diffusion ELBO in Eq. (22) and re-expresses it under the ASM-based Gaussian transitions. By substituting the mean-shifted forward process into the ELBO and simplifying the KL terms, the bound reduces to the training objective in Eq. (49). Thus, Eq. (49) is directly derived from a valid evidence lower bound on the data log-likelihood, and minimizing Eq. (49) corresponds to maximizing this ELBO up to constant factors. In this sense, ASM modifies the **bias** of the diffusion dynamics by steering them along a physically meaningful haze-formation trajectory, and preserves the underlying probabilistic formulation and the ELBO-based convergence analysis. Empirically, our ablation on different diffusion priors in Table 4 confirms that the ASM-integrated HNDiff prior improves PSNR much more than a noise-only diffusion prior under the same training protocol.

---

> ### Author Response · Authors · 2025-11-20
> **Rebuttal by Authors (Weakness 3, Question 1 to 2)**
>
> ### **Weakness 3-1: This work appears to be just an implementation and improvement of latent diffusion under an atmospheric scattering model.**
>
> As discussed in Section 2.2, most existing methods utilizing latent diffusion as a prior simply adopt a conventional diffusion model, which adds zero-mean Gaussian noise until the image becomes pure Gaussian noise, without explicitly incorporating the underlying physics. In contrast, HNDiff tightly integrates the Atmospheric Scattering Model (ASM) into the diffusion process by shifting the mean of the Gaussian corruption, yielding an ASM-guided haze–noise diffusion with a rigorous mathematical formulation and convergence analysis (Appendix A.2, Eq.(22)–(49)). Moreover, the Haze-Aware Noise Scheduler (HANS) further modulates the noise level according to local haze density, aligning stochastic corruption with the spatially varying haze pattern predicted by ASM. Empirically, Table 4 further shows that this physics-guided diffusion prior delivers significantly larger gains than other diffusion-based priors under the same backbone and training protocol, indicating that HNDiff goes beyond a straightforward implementation of an existing latent diffusion pattern.
>
> ### **Weakness 3-2 & Question 1: Using the potential diffusion model seems to obscure the advantage of interpretability. How does the latent diffusion representation compare to full-image diffusion in terms of interpretability and potential loss of fine spatial information?**
>
> As discussed in Lines 298-302, applying diffusion directly in the image space often leads to substantial computational overhead, degradation of distortion-based fidelity, and slower, less stable convergence. Our choice to perform diffusion in a compact latent space mitigates these issues while still encoding haze-related structure in a controllable way.
>
> In our design, the latent representation is specifically intended to capture low-resolution, global haze cues. Since our encoder is a CNN-based network, the latent feature map preserves the spatial layout and local spatial relationships of the input image, so haze-related patterns remain organized in a geometrically meaningful way on the $H/4 \times W/4$ grid. The diffusion states in latent space can therefore be interpreted as progressively refined estimates of how haze distribution and veil strength vary across the scene. Fine spatial details and high-frequency textures are deliberately left to the dehazing backbone, which receives this latent prior and uses it to guide its decoder features. In this way, the latent diffusion provides an interpretable global prior, while the backbone is responsible for reconstructing local structures.
>
> Empirically, the visualization of $Z_{4 \rightarrow 0}$ in Figure 5 shows a clear progression from hazy to clean in the latent maps, supporting our interpretation that the latent diffusion states encode meaningful, haze-related structure rather than losing interpretability.
>
> ### **Question 2: Is the improvement mainly due to the physical prior or the additional parameters introduced by the latent modules and FGM?**
>
> We address this question from two perspectives: enlarged baseline variants and the parameters introduced by the latent modules and FGM.
>
> **Enlarged baseline variants.**
> To disentangle the effect of additional parameters, we have compared our proposed HNDiff with two enlarged baseline variants in Table 5 of the paper. Both variants substantially increase the parameter count ($8.40$M and $8.28$M) but yield only marginal PSNR gains over the baselines ($+0.01$ dB and $+0.15$ dB). In contrast, HNDiff delivers significantly larger performance gains ($+0.53$ dB) with fewer parameters ($7.82$M), indicating that the improvement stems from the proposed physics-guided diffusion prior rather than simply adding more parameters.
>
> **Parameters introduced by latent modules and FGM.**
> To isolate the effect of the additional parameters introduced by the latent modules and FGM, Table 4 compares different diffusion priors under the same configuration (latent module, FGM, three-stage training, and dehazing backbone). Alternative priors yield at most about $0.1$ dB PSNR improvement, whereas our HNDiff prior provides a $0.53$ dB gain, indicating that the main performance improvement comes from the proposed haze–noise diffusion design rather than merely the extra parameters introduced by the latent modules and FGM.

---

> ### Author Response · Authors · 2025-11-20
> **Rebuttal by Authors (Question 3 to 4)**
>
> ### **Question 3: Whether the haze-noise diffusion pattern designed by this paper can be applied to image-to-image diffusion models rather than latent diffusion models.**
>
> We thank the reviewer for this interesting question. The proposed HNDiff is not restricted to latent models. The formulation in Section 3.1 is defined in image space, and the latent version simply applies the same haze–noise diffusion to an encoded representation. In the paper, we adopt the latent formulation primarily to avoid the substantial computational overhead, fidelity degradation, and slower convergence commonly observed when applying image-space diffusion to restoration tasks (Lines 298-302), while still encoding ASM-guided haze structure.
>
> Following the reviewer’s question, we implement an image-space variant, HNDiff (Image), and compare it with the latent version, HNDiff (Latent), and the baseline FocalNet on the real-world RW2AH dataset. Both HNDiff variants use the same U-Net as the haze/noise estimator, while HNDiff (Latent) further employs FocalNet as the dehazing backbone. As shown in Table C, HNDiff (Latent) achieves the best dehazing results ($21.52$ dB PSNR, $0.6254$ SSIM) with only a minimal additional FLOPs overhead (36.38G), confirming its advantage in content fidelity, and both HNDiff variants outperform the baseline FocalNet. These results demonstrate that HNDiff is applicable in both image and latent spaces, with the latent version offering better fidelity and efficiency. We have added this comparison and qualitative visualizations in Section 4.3 of the revised paper as Table 7 and Figure 6.
>
> **Table C.** Comparison of applying HNDiff in image space and latent space on the RW²AH test set.
>
> | Metric | FocalNet | HNDiff (Image) | HNDiff (Latent) |
> |--------|----------|----------------|------------------|
> | **PSNR (dB)** | 21.18 | 21.37 | **21.52** |
> | **SSIM** | 0.5970 | 0.6166 | **0.6254** |
> | **FLOPs (G)** | 30.35 | 65.59 | 36.38|
> ### **Question 4: Results on RTTS dataset.**
>
> We thank the reviewer for this comment. Following the suggestion, we have conducted additional experiments on the RTTS dataset. Our models, pretrained on the NH-HAZE dataset, are directly evaluated on RTTS to assess out-of-domain generalization. As shown in Table D, the HNDiff-enhanced models consistently achieve better results (indicated by lower BRISQUE and NIQE scores) than their corresponding baselines, thereby demonstrating better perceptual quality on this real-world benchmark. These results on RTTS have been incorporated into Appendix A.4 in the revised paper.
>
> **Table D.** Results on RTTS, where the proposed HNDiff is applied to three baselines: FocalNet, ConvIR, and SGDN.
>
> | Metric        | FocalNet | HNDiff (FocalNet) | ConvIR  | HNDiff (ConvIR) | SGDN    | HNDiff (SGDN) |
> |---------------|----------|-------------------|---------|------------------|---------|----------------|
> | **BRISQUE ↓** | 35.9789  | **29.8278**       | 36.5144 | **34.9876**      | 34.6808 | **32.5563**    |
> | **NIQE ↓**    | 4.3392   | **4.2867**        | 4.3992  | **4.2665**       | 4.8426  | **4.6343**     |

---

> ### Author Response · Authors · 2025-11-25
>
> Dear reviewer,
>
> Thank you for the comments on our paper.
>
> We have submitted the responses to your comments. We would like to confirm whether our responses sufficiently address your concerns. Please let us know if you have additional questions so that we can address them during the discussion period.
>
> Thank you!
>
> Best,
>
> Authors

---

> ### Comment · Reviewer_HBqs · 2025-11-26
>
> Thanks to the authors' response,
>
> These comments have addressed part of my concerns. However, the results on the RTTS dataset show that its performance in real-world scenarios are limited compared to several earlier ASM-based methods such as [1,2,3,5] (But if HNDiff can also achieve improvement on some of these methods, that would be a very good thing). In addition, the manuscript lacks sufficient discussion of related ASM methods [1,2,3,4,5,6], even though the core contributions of this work are built upon ASM.
>
> I also recommend that the authors report FADE and NIMA results on RTTS (preferably with models trained on OTS), as well as object detection results on RTTS, to further validate performance in real-world tasks.
>
> If the authors can provide these improvements, I'm glad to increase my rating.
>
> By the way, it is recommended to start the appendix on a new page.
>
> [1] HazeFlow, ICCV 2025
>
> [2] CORUN, NeurIPS 2024
>
> [3] RIDCP, CVPR 2023,
>
> [4] D4, CVPR 2022
>
> [5] PSD, CVPR 2021
>
> [6] DAD, CVPR 2020

---

> > ### Author Response · Authors · 2025-11-28
> > **Response to follow-up comments**
> >
> > ### **Follow-up Comments: The results on the RTTS dataset show that its performance in real-world scenarios are limited compared to several earlier ASM-based methods such as [1,2,3,5] (But if HNDiff can also achieve improvement on some of these methods, that would be a very good thing). In addition, the manuscript lacks sufficient discussion of related ASM methods [1,2,3,4,5,6], even though the core contributions of this work are built upon ASM. I also recommend that the authors report FADE and NIMA results on RTTS (preferably with models trained on OTS), as well as object detection results on RTTS, to further validate performance in real-world tasks.**
> >
> > We thank the reviewer for the insightful comments and helpful suggestions. Below, we address the concerns in detail.
> >
> > Following the reviewer’s suggestion, we adopt a representative ASM-based method, RIDCP [3], as the backbone and train the pipeline on the OTS dataset. In the current rebuttal, we update the results for RIDCP only, as other ASM-based methods [1,2,5] additionally rely on the URHI dataset (4,807 images) and incorporate extra pseudo-label generation and fine-tuning stages. Thus, it would take much more time to fully reproduce these training pipelines. These experiments are currently in progress; for now, we report results based on RIDCP, which serves as a representative and reproducible baseline.
> >
> > As shown in Table F, integrating HNDiff brings consistent improvements over RIDCP on RTTS across FADE, NIMA, and BRISQUE. To further validate practical benefits, we conduct a real-world downstream task evaluation by reporting class-wise mAP@50 on RTTS using a pretrained YOLOv3 detector (Table G). HNDiff (RIDCP) achieves higher mAP scores for all categories than RIDCP alone, demonstrating clear benefits for downstream perception tasks. Overall, these results show that the priors generated by HNDiff can also substantially enhance existing ASM-based backbones in real-world scenarios. Additionally, the qualitative results in Figure 22 of the revised paper further illustrate that HNDiff produces more natural and cleaner dehazing results.
> >
> >
> > **Table F. Quantitative comparison of RIDCP and HNDiff (RIDCP) on RTTS.**
> > | Method            | FADE ↓ | NIMA ↑ | BRISQUE ↓ |
> > |-------------------|--------|--------|------------|
> > | RIDCP             | 0.944  | 4.97   | 17.29      |
> > | **HNDiff (RIDCP)** | **0.417** | **5.08** | **16.09** |
> >
> >
> > **Table D: Object detection performance (mAP@50) on RTTS using a pretrained YOLOv3.**
> > | Method          | person | bicycle | car   | motorcycle | bus   | mean  |
> > |-----------------|--------|---------|-------|------------|-------|-------|
> > | Hazy Image      | 0.662  | 0.425   | 0.581 | 0.376      | 0.299 | 0.469 |
> > | RIDCP           | 0.669  | 0.444   | 0.611 | 0.448      | 0.341 | 0.503 |
> > | **HNDiff (RIDCP)** | **0.677** | **0.454** | **0.629** | **0.452** | **0.361** | **0.515** |
> >
> > Following the reviewer’s suggestion, we have added a discussion of prior ASM-based dehazing methods in Section 2.1 of the revised paper. We also updated the layout so that the appendix now starts on a new page. We hope that these additional experiments and revisions adequately address the reviewer’s concerns and strengthen the contribution of our work. We are happy to provide any further details if needed. Thank you for your time and consideration.
> >
> > [1] HazeFlow, ICCV 2025
> >
> > [2] CORUN, NeurIPS 2024
> >
> > [3] RIDCP, CVPR 2023,
> >
> > [4] D4, CVPR 2022
> >
> > [5] PSD, CVPR 2021
> >
> > [6] DAD, CVPR 2020

---

### Official Review · Reviewer_JRmV · 2025-10-29

**Soundness:** 3
**Presentation:** 2
**Contribution:** 2
**Rating:** 4
**Confidence:** 4

**Summary:**

The paper introduces Haze-Noise Diffusion (HNDiff), a diffusion-based image dehazing framework that integrates the atmospheric scattering model as a physical prior. Unlike conventional methods that add only Gaussian noise, HNDiff employs a haze-aware noise scheduler that adapts noise levels based on haze density. The authors also propose Latent HNDiff, which leverages clean latent priors to enhance existing dehazing networks.

**Strengths:**

1. HNDiff integrates the physical model into the diffusion process, mimicking the formation of haze by highlighting its spatially varying characteristics.
2. The latent formulation reduces computational cost while improving restoration accuracy and visual fidelity.

**Weaknesses:**

1. The paper introduces the Haze-Noise Diffusion process, claiming to incorporate the physical mechanism of haze formation as an inductive bias for improved dehazing. While the methodology is detailed, the authors fail to clearly justify the necessity of this inclusion. Specifically, the paper lacks a **clear, intuitive explanation** to help the reader understand why this specific combination of haze and noise addition in Eq.(2), is more effective.
2. The manuscript introduces several distinct components, including HNDiff framework, Haze-Aware Noise Scheduler, and Latent HNDiff with FGM, yet it does not establish a clear hierarchical structure or narrative focus among them. This lack of prioritization weakens the overall coherence of the work, making it difficult for readers to discern which element represents the paper’s primary innovation or how these components collectively address a unified research question.

**Questions:**

1. What does HANS stand for in Figure 2?
2. The manuscript claims that the proposed model achieves effective dehazing within only four diffusion steps. Given such an extremely limited number of iterations, what meaningful dehazing features are actually learned? How these few steps capture haze-related representations and improve restoration quality?

---

> ### Author Response · Authors · 2025-11-20
> **Rebuttal by Authors**
>
> We thank the reviewer for recognizing the physical grounding of our approach and the practicality of the latent design. We greatly appreciate these encouraging comments. We address each concern in detail below and have revised the paper accordingly.
>
> ### **Weakness 1: the paper lacks a clear, intuitive explanation to help the reader understand why this specific combination of haze and noise addition in Eq.(2), is more effective.**
>
> As discussed in Section 3.1, haze is **structured and spatially varying** based on the Atmospheric Scattering Model (ASM). In contrast, conventional diffusion models define the forward process by repeatedly adding zero-mean Gaussian noise until the image becomes pure noise, disregarding this spatially varying physical process of haze formation. As a result, the conventional trajectory ignores how haze is physically formed. It follows that the reverse process must reconstruct the clean image from purely random noise rather than from a physically meaningful hazy state.
>
> Motivated by this discrepancy, our Haze-Noise Diffusion (HNDiff) model in Eq. (2) explicitly incorporates ASM as a **mean shift** of the Gaussian transitions: the mean of $q(I_t \mid I_{t-1})$ follows the ASM-based clean-to-hazy evolution (Eq. (3)), while Gaussian noise is injected around this trajectory. This design forces the forward process move along a physically grounded clean-to-hazy manifold, and the reverse process inverts this ASM-guided evolution by jointly removing haze and noise. Consequently, the haze+noise formulation in Eq. (2) offers a stronger inductive bias than (i) noise-only diffusion, which ignores haze physics, and (ii) purely deterministic haze estimation, which lacks stochastic modeling capacity in heavily degraded regions.
>
> We have added this intuitive explanation around Eq.(2) in Section 3.1 of the revised paper.
>
> ### **Weakness 2: Hierarchical structure of the proposed method with narrative focus among the proposed components, including Haze-Aware Noise Scheduler and Latent HNDiff with FGM.**
>
> We appreciate this constructive comment. Following this suggestion, we have restructured the first paragraph of Section 3 in the revised paper to present our method using a **top-down hierarchical structure.** This revised presentation first highlights our core innovation, the HNDiff diffusion algorithm, offering an overview through the network pipeline (Figure 2). It then describes the roles and relationships of the individual components in HNDiff, and clarifies how these components collectively achieve the unified goal of providing a physically grounded diffusion prior for image dehazing.
>
> For details, please refer to the first paragraph of Section 3 in the revised paper. Here is a summary: our method is organized in a top–down hierarchy, with the new diffusion algorithm HNDiff as the primary contribution. Within HNDiff, the Haze-Noise Diffusion, the Haze-Aware Noise Scheduler (HANS), and the Dehazing–Denoising Process form the core components of the framework. For practical deployment, we further build Latent HNDiff with FGM on top of HNDiff to improve existing dehazing backbones in a plug-and-play manner.
>
> ### **Question 1: What does HANS stand for in Figure 2?**
>
> We thank the reviewer for pointing this out. HANS denotes the Haze-aware Noise Scheduler in Figure 2. We have introduced and defined this abbreviation at the beginning of Section 3 in the revised paper.
>
> ### **Question 2: Questions about the dehazing features and the dehazing performance within few steps.**
>
> The dehazing features represent the haze residuals that should be removed at each step. As shown in Figure 5, these haze patterns are gradually eliminated through the iterative dehazing–denoising process. Since our diffusion model operates as a prior generator in the latent space, serving as an auxiliary module to enhance the dehazing backbones, it can improve restoration quality within only a few steps, rather than requiring numerous iterations typically needed in traditional image-space diffusion models. This efficiency is consistent with findings in recent latent-diffusion-based methods for image restoration [A, B], which required only eight denoising steps in [A] and four denoising steps in [B].
>
> **Reference**
> #### [A] Zheng Chen, Yulun Zhang, Ding Liu, Jinjin Gu, Linghe Kong, Xin Yuan, et al. Hierarchical integration diffusion model for realistic image deblurring. In NeurIPS, 2023.
> #### [B] Bin Xia, Yulun Zhang, Shiyin Wang, Yitong Wang, Xinglong Wu, Yapeng Tian, Wenming Yang, and Luc Van Gool. Diffir: Efficient diffusion model for image restoration. In ICCV, 2023.

---

> ### Author Response · Authors · 2025-11-25
>
> Dear reviewer,
>
> Thank you for the comments on our paper.
>
> We have submitted the responses to your comments. We would like to confirm whether our responses sufficiently address your concerns. Please let us know if you have additional questions so that we can address them during the discussion period.
>
> Thank you!
>
> Best,
>
> Authors

---

### Official Review · Reviewer_96ea · 2025-10-29

**Soundness:** 3
**Presentation:** 3
**Contribution:** 4
**Rating:** 8
**Confidence:** 4

**Summary:**

In this paper, the authors propose the HNDiff (Haze-Noise Diffusion) framework, which integrates the Atmospheric Scattering Model (ASM) as an inductive bias into the diffusion process. Through a bidirectional pipeline of "joint haze-noise diffusion" and "physically consistent dehazing-denoising", it addresses the issues of low restoration fidelity in traditional diffusion-based dehazing methods—specifically, their neglect of the physical principles of haze formation and reliance on pure Gaussian noise.   Additionally, the proposed Latent HNDiff can serve as a prior generation network, enabling flexible integration with existing dehazing backbones. Experiments demonstrate that HNDiff consistently enhances the performance of three mainstream dehazing models (FocalNet, ConvIR, and SGDN) across five benchmark datasets.   Moreover, it outperforms schemes that merely scale up network size while requiring fewer parameters and lower computational complexity, verifying its effectiveness and generalization.

**Strengths:**

Physical grounding: Embedding ASM into diffusion is a critical strength—unlike conventional diffusion methods that ignore haze’s structured, spatially varying nature, HNDiff’s forward/reverse processes align with how haze forms in the real world.  This ensures more realistic restorations and better generalization to real-world haze.

Latent HNDiff’s ability to enhance existing dehazing backbones (FocalNet, ConvIR, SGDN) without reengineering them makes it a valuable tool for researchers and engineers. The FGM module’s lightweight design (pooling + MLP) ensures minimal overhead, and the three-stage training balances stability and performance.

The paper’s results are convincing across metrics and datasets.

**Weaknesses:**

The paper mentions the haze estimator uses a simplified U-Net. However, it does not provide: (a) the exact architecture of the U-Net (e.g., number of layers, channel counts, skip connections); (b) the loss function used to train the estimator; (c) The performance of the haze estimator in extremely dense haze scenarios.

The paper has verified the consistency between HNDIFF and the Atmospheric Scattering Model (ASM) under uniform haze scenarios. However, haze in real-world environments is often non-uniformly distributed (e.g., gradient haze where density increases with depth). It remains unclear how the current haze-aware noise scheduler adapts to such scenarios.

**Questions:**

Does the implicit haze residual modeling (via continuous accumulation) effectively capture non-uniform haze density variations?
How do the estimators avoid mistaking image details (e.g., leaf textures, building edges) for haze residuals? For instance, a dense forest may have texture densities similar to light haze.

---

> ### Author Response · Authors · 2025-11-20
> **Rebuttal by Authors**
>
> We thank the reviewer for recognizing the physical grounding, practicality, and experimental strength of our work. We greatly appreciate these positive and encouraging comments. We address each concern in detail below and have revised the paper accordingly.
>
> ### **Weakness 1-1: What is the exact architecture of the U-Net (e.g., number of layers, channel counts, skip connections)?**
>
> As mentioned in Section 4.1, our haze estimator adopts the simplified U-Net design proposed by Liu et al. [A]. Specifically, it is a 4-stage U-Net that starts with 32 feature channels at the first scale and increases the channel size to 64, 128, and 128 at the subsequent scales. Each encoder and decoder stage contains two residual blocks with time conditioning. Standard U-shaped skip connections are employed by concatenating encoder and decoder features at the same resolution. These architectural details have been added to Appendix A.8 in the revised manuscript. We will release the full source code upon paper acceptance.
>
> ### **Weakness 1-2: What is the loss function used to train the estimator?**
>
> In Stage 2, we train both the haze and noise estimators using the latent-prior loss $L_{prior} = |Z_0 - Z_{gt}|$, as stated in Line 344 of the revised paper. This loss supervises the final latent output $Z_0$ to match the ground-truth prior $Z_{\text{gt}}$. More details on this objective and its role in our formulation are provided in Lines 340-347 and Lines 878-888 (Appendix A.2) of the revised paper.
>
> ### **Weakness 1-3: Performance of the haze estimator in dense haze scenarios?**
>
> We have explicitly evaluated our method on the Dense-HAZE dataset, which is specifically designed to capture extremely dense haze conditions. As reported in Table 1, our HNDiff-enhanced models consistently improve all three backbones and achieve state-of-the-art performance under this challenging setting. These findings indicate that the proposed haze estimator remains effective and robust even in extremely dense haze conditions.
>
> ### **Weakness 2 & Question 1-1: Non-uniform haze scenarios.**
>
> We would like to first emphasize that HNDiff is **not** restricted to uniform haze. As defined in Eq. (1) (Section 3.1), a transmission map $\tau \in \mathbb{R}^{H \times W \times 1}$ in the ASM is modeled by the corresponding scattering coefficient $\sigma$ and depth map $d$, both also defined per pixel ($\sigma, d \in \mathbb{R}^{H \times W \times 1}$). Thus, the formulation inherently supports **spatially varying** (non-uniform) haze, rather than assuming the uniform case. Building on this formulation, we define the haze residual as a spatial map $1 - \tau_t\in \mathbb{R}^{H \times W \times 1}$, so each pixel has its own haze level. The proposed Haze-Aware Noise Scheduler then uses this per-pixel residual to modulate the noise scale, allowing both HNDiff and HANS to adapt to non-uniform haze distributions in the forward and reverse processes.
>
> From an empirical standpoint, HNDiff is evaluated on several datasets with non-uniform or real-world haze, including NH-HAZE, Dense-HAZE, O-HAZE, and RW2AH, all of which exhibit substantial spatial variation in haze density. As shown in Table 1, our method consistently surpasses strong baselines on these benchmarks, indicating that the implicit haze-residual modeling and the haze-aware scheduler effectively handle non-uniform haze in practice.
>
> ### **Question 1-2: Concerns regarding how the haze estimator avoids confusion between image details and haze residuals.**
>
> To respond to the mentioned concern, we highlight that the design of the haze estimator explicitly reduces the risk of confusing image structures with haze.
> As shown in Lines 290-292, the haze estimator predicts the complement of the transmission,
> $
>   1 - e^{-\alpha_t o^\theta(I_t, I_H, t)},
> $
> which is designed to represent only the haze component rather than image details. By modeling the attenuation, the haze estimator naturally focuses on extracting haze-specific features instead of scene texture.
>
> In addition, our three-stage training pipeline with a latent prior loss further discourages confusing image details with haze. In Stage 2, HNDiff is optimized using the latent prior loss, so if textures (e.g., leaves or building edges) are mistakenly treated as haze, the resulting latent prior deviates from the clean-image ground truth and increases this loss. This drives the model to correct such mistakes and avoid suppressing genuine image details. At last, in Stage 3, joint fine-tuning of the whole framework with image-level dehazing loss further penalizes the removal of true image content and guides the estimators to concentrate on actual haze components rather than image textures. Altogether, the formulation and training strategy effectively prevent confusion between haze and image details.

---

> ### Author Response · Authors · 2025-11-25
>
> Dear reviewer,
>
> Thank you for the comments on our paper.
>
> We have submitted the responses to your comments. We would like to confirm whether our responses sufficiently addresses your concerns. Please let us know if you have additional questions so that we can address them during the discussion period.
>
> Thank you!
>
> Best,
>
> Authors

---

### Official Review · Reviewer_ud8n · 2025-10-31

**Soundness:** 2
**Presentation:** 2
**Contribution:** 2
**Rating:** 4
**Confidence:** 4

**Summary:**

In this paper, the authors proposed a new framework, termed HNDiff, for image dehazing task. Its latent version can be integrated into existing dehazing networks to boost performance. The experimental results show that it can be successfully applied into classical models (e.g., FocalNet, ConvIR, and SGDN).

**Strengths:**

In summary, HNDiff represents an advancement in diffusion-based image dehazing by embedding physical principles into the diffusion process. The theoretical derivation looks highly rigorous.

**Weaknesses:**

1. The gt prior $Z_{gt}$ is generated from the hazy image $I_H$ and its clean counterpart $I_0$. What is the function of  $I_H$ here? From my perspective, $I_H$ will only bring about negative impacts.
2. The visualization results in Figure 5 are not reliable. The authors merely selected the first channel, which is prone to be biased. A relatively more reasonable approach would be to average each latent over the channel dimension or adopt other statistically meaningful methods. By the way, the dimensions of $Z_{0:4}$ are not identical to $I_H$. If any resize operations are applied here, clearly explanations should be provided.
3. The entire training process requires three stages, which is quite complex and may cause robustness issues. During the stage 2, only a latent-prior loss is employed? The estimated haze and noise are not supervised?
4. The idea of haze-aware noise scheduler (dynamically adjust the noise level according to haze density) has been explored before.
5. The HANS in Figure 2 is not defined before it is used.
6. Experiments on real hazy images (e.g., RTTS) are missing. In Table 1, SOTS-outdoor is more suggested than SOTS-indoor. The visual results on other models besides FocalNet, ConvIR, and SGDN are not provided. The visual result of HNDiff in Figure 13 contains some artifacts.

**Questions:**

Please refer to the weaknesses part.

---

> ### Author Response · Authors · 2025-11-20
> **Rebuttal by Authors (Weakness 1 to 3)**
>
> We thank the reviewer for recognizing the contribution of HNDiff in embedding physical principles into the diffusion process and for the positive assessment of our theoretical derivation. We greatly appreciate these encouraging comments. We address each concern in detail below and revise the paper accordingly.
>
> ### **Weakness 1: What is the function of $I_H$ here? From my perspective, $I_H$ will only bring about negative impacts.**
>
> Using both the hazy image $I_H$ and its clean counterpart $I_0$ to predict the ground-truth prior $Z_{gt}$ allows the encoder to explore the **correspondence** between the hazy and clean images, which significantly aids in prior prediction. To verify this statement, we compare two variants of $Z_{gt}$ injected into a dehazing network and evaluate their Stage 1 dehazing performance: (a) $Z_{gt}$ generated using $I_0$ only, i.e., $Z_{gt} = \mathrm{IE}\bigl(I_0\bigr)
> $, and (b) $Z_{gt}$ generated using both $I_H$ and $I_0$, i.e., $Z_{gt} = \mathrm{IE}\bigl(\mathrm{Concat}(I_H, I_0)\bigr)$, which is the setting used in our paper. As shown in Table A, setting (b) achieves better dehazing performance on the NH-HAZE dataset with FocalNet, confirming that incorporating $I_H$ provides helpful haze-related cues essential for generating a stronger ground-truth prior.
>
> **Table A.** Stage 1 dehazing performance of FocalNet on NH-HAZE using $Z_{gt}$ generated using (a) $I_0$ only and (b) the concatenation of $I_H$ and $I_0$.
> | Variant | Encoder (IE) Input      | PSNR (dB) |
> |--------|-------------------------|----------:|
> | (a)    | $I_0$                 |    20.74  |
> | (b)    | $\text{Concat}(I_H, I_0)$ |    21.36  |
>
> ### **Weakness 2: The visualization results in Figure 5 are not reliable. A relatively more reasonable approach would be to average each latent over the channel dimension or adopt other statistically meaningful methods. By the way, the dimensions of $Z_{0:4}$ are not identical to $I_H$. If any resize operations are applied here, clearly explanations should be provided.**
>
> Thank you for this insightful suggestion. In Figure 5 of the revised paper, we follow your recommendation by averaging latent features over the channel dimension for visualization, which yields a more robust representation. We have also clarified the spatial resolution of Figure 5 in Line 505 of the revised paper. Since $Z_{0:4} \in \mathbb{R}^{\frac{H}{4} \times \frac{W}{4} \times C}$, we bilinearly downsample $I_H$ to match the size of $Z_{0:4}$ for visualization purposes only.
>
> ### **Weakness 3-1: Three-stage training process is quite complex.**
>
> The three-stage training scheme has been widely adopted in recent latent diffusion-based image restoration methods [A, B, C] to generate the ground-truth prior. It has, hence, been verified to be effective and stable.
>
> ### **Weakness 3-2: During the stage 2, only a latent-prior loss is employed? The estimated haze and noise are not supervised?**
>
> Since ground-truth haze residuals are unavailable for intermediate diffusion steps, Stage 2 relies solely on the latent-prior loss $\mathcal{L}{\text{prior}}$ to optimize the image encoder, haze estimator, and noise estimator. We supervise only the final latent $Z_0$, allowing gradients to propagate through the entire diffusion trajectory, as described in Lines 340-347 and Lines 878-888 (Appendix A.2) of the revised paper. In our setting, $Z_0$ is reconstructed from $Z_T$ by repeatedly applying the haze and noise estimators; therefore, these estimators are implicitly supervised through the alignment of the predicted $Z_0$ with the ground-truth prior $Z{\text{gt}}$. This training strategy is consistent with recent diffusion-based image restoration approaches [A, B, C], where supervision, likewise, on the final reconstruction is sufficient to train all the intermediate denoising steps.
>
> **Reference**
> #### [A] Chen Rao, Guangyuan Li, Zehua Lan, Jiakai Sun, Junsheng Luan, Wei Xing, Lei Zhao, Huaizhong Lin, Jianfeng Dong, and Dalong Zhang. Rethinking video deblurring with wavelet-aware dynamic transformer and diffusion model. In ECCV, 2024.
> #### [B] Zheng Chen, Yulun Zhang, Ding Liu, Jinjin Gu, Linghe Kong, Xin Yuan, et al. Hierarchical integration diffusion model for realistic image deblurring. In NeurIPS, 2023.
> #### [C] Bin Xia, Yulun Zhang, Shiyin Wang, Yitong Wang, Xinglong Wu, Yapeng Tian, Wenming Yang, and Luc Van Gool. Diffir: Efficient diffusion model for image restoration. In ICCV, 2023.

---

> ### Author Response · Authors · 2025-11-20
> **Rebuttal by Authors (Weakness 4 to 6)**
>
> ### **Weakness 4: The idea of haze-aware noise scheduler (dynamically adjust the noise level according to haze density) has been explored before.**
>
> To the best of our knowledge, existing related diffusion-based methods [D, E, F] utilize haze density or transmission only as auxiliary guidance for the denoising model, while still following the standard diffusion formulation, i.e., adding zero-mean Gaussian noise and gradually driving the image distribution toward pure noise.
>
> In contrast, we reformulate the forward process by introducing a continuous haze diffusion process and integrating it directly into the noise diffusion pipeline. This allows the noise level to be modulated by the local haze density, which changes the underlying generative process rather than merely providing external guidance to the denoiser. For clarity and mathematical completeness, we include the full ELBO derivation and optimization analysis of our formulation in Appendix A.2.
>
> We appreciate the reviewer's comment and would be happy to further discuss this point. If there exist works that more closely resemble our haze-aware noise scheduling mechanism, we would be grateful if the reviewer could share them.
>
> ### **Weakness 5: The HANS in Figure 2 is not defined.**
>
> We thank the reviewer for pointing this out. In the revised paper, we have explicitly introduced and defined the Haze-aware Noise Scheduler (HANS) at the beginning of Section 3 before its first appearance in Figure 2.
>
> ### **Weakness 6-1: Experiments on real hazy images (e.g., RTTS) are missing.**
>
> We would like to clarify that our main results in Table 1 of the paper already incorporate performance evaluations on multiple real-world hazy datasets, including NH-HAZE, O-HAZE, and Dense-HAZE collected using a fog machine, as well as RW$^2$AH, which contains real haze captured by fixed webcams.
>
> Following the reviewer’s suggestion, we additionally conducted further experiments on the RTTS dataset, which consists of real-world hazy images without corresponding clean images. For the evaluation, we use models pretrained on the NH-HAZE dataset. As shown in Table B, our method consistently achieves better scores (lower BRISQUE and NIQE) than the baselines, confirming that the proposed approach is effective on the real-world RTTS dataset as well. These new results have been added to Appendix A.4 of the revised paper.
>
> **Table B.** Results on RTTS, where the proposed HNDiff is applied to three baselines: FocalNet, ConvIR, and SGDN.
>
> | Metric        | FocalNet | HNDiff (FocalNet) | ConvIR  | HNDiff (ConvIR) | SGDN    | HNDiff (SGDN) |
> |---------------|----------|-------------------|---------|------------------|---------|----------------|
> | **BRISQUE ↓** | 35.9789 | **29.8278**    | 36.5144 | **34.9876** | 34.6808 | **32.5563**    |
> | **NIQE ↓**  | 4.3392| **4.2867** | 4.3992| **4.2665** | 4.8426 | **4.6343**   |
>
> ### **Weakness 6-2: SOTS-outdoor is more suggested than SOTS-indoor.**
>
> We thank the reviewer for this suggestion. We are currently conducting additional experiments on SOTS-outdoor with the same settings and will include the corresponding results in the revised manuscript before the end of the rebuttal period.
>
> ### **Weakness 6-3: The visual results on other models besides FocalNet, ConvIR, and SGDN are not provided.**
>
> We primarily present visual comparisons for the leading state-of-the-art methods: FocalNet, ConvIR, SGDN, and their HNDiff-enhanced versions. In response to the reviewer’s request, we have included visual results for the remaining competing methods listed in Table 1, now provided in Appendix A.7 of the revised paper.
>
> ### **Weakness 6-4: The visual result of HNDiff in Figure 13 contains some artifacts.**
>
> Figure 13 presents particularly challenging real-world hazy images that even strong, state-of-the-art baselines struggle to recover effectively. For example, the baseline ConvIR misinterprets the bright sky region as heavy haze, resulting in a pronounced dark artifact in the upper part of the image. In contrast, our HNDiff-enhanced model alleviates this failure case, better preserving the sky appearance and the overall scene structure. Overall, HNDiff consistently improves the baseline models, yielding visibly cleaner dehazing results and avoiding the severe artifacts observed in the baselines.
>
> **Reference**
> #### [D] Ling, Zhang, Bai, Wenxu, and Xiao, Chunxia. Density‐aware diffusion model for efficient image dehazing. In Computer Graphics Forum, 2024.
> #### [E] Ruicheng Zhang, Puxin Yan, Zeyu Zhang, Yicheng Chang, Hongyi Chen, and Zhi Jin. Rpd-diff: region-adaptive physics-guided diffusion model for visibility enhancement under dense and non-uniform haze. arXiv preprint, 2025.
> #### [F] Shibai Yin, Yiwei Shi, Yibin Wang, and Yee-Hong Yang. When aware haze density meets diffusion model for synthetic-to-real dehazing. IEEE Transactions on Circuits and Systems for Video Technology, 2025.

---

> > ### Author Response · Authors · 2025-11-28
> > **Weakness 6-1: Experiments on real hazy images (e.g., RTTS) are missing. (more experimental results.)**
> >
> > We additionally conduct experiments on the real-world RTTS dataset to verify that our method remains effective and achieves state-of-the-art performance in real-world scenarios. Since RIDCP is a state-of-the-art dehazing method on RTTS, we adopt it as our backbone and build HNDiff on top of RIDCP. As shown in Table H, HNDiff (RIDCP) consistently improves over the RIDCP backbone across all no-reference metrics on RTTS, reducing FADE from 0.944 to 0.417, increasing NIMA from 4.97 to 5.08, and lowering BRISQUE from 17.29 to 16.09. Furthermore, in a downstream object detection evaluation using a pretrained YOLOv3 (Table I), HNDiff (RIDCP) yields higher mAP@50 for all categories. These results demonstrate that the priors generated by HNDiff significantly enhance a state-of-the-art dehazing backbone and lead to state-of-the-art performance on RTTS in real-world conditions.
> >
> > **Table H. Quantitative comparison of RIDCP and HNDiff (RIDCP) on RTTS.**
> > | Method            | FADE ↓ | NIMA ↑ | BRISQUE ↓ |
> > |-------------------|--------|--------|------------|
> > | RIDCP             | 0.944  | 4.97   | 17.29      |
> > | **HNDiff (RIDCP)** | **0.417** | **5.08** | **16.09** |
> >
> > **Table I: Object detection performance (mAP@50) on RTTS using a pretrained YOLOv3.**
> > | Method          | person | bicycle | car   | motorcycle | bus   | mean  |
> > |-----------------|--------|---------|-------|------------|-------|-------|
> > | Hazy Image      | 0.662  | 0.425   | 0.581 | 0.376      | 0.299 | 0.469 |
> > | RIDCP           | 0.669  | 0.444   | 0.611 | 0.448      | 0.341 | 0.503 |
> > | **HNDiff (RIDCP)** | **0.677** | **0.454** | **0.629** | **0.452** | **0.361** | **0.515** |

---

> ### Author Response · Authors · 2025-11-25
> **Rebuttal by Authors (Weakness 6-2)**
>
> ### **Weakness 6-2: SOTS-outdoor is more suggested than SOTS-indoor.**
>
> Following the reviewer’s suggestion, we evaluate on the SOTS-Outdoor dataset. Integrating HNDiff as a plug-in prior into three representative backbones yields consistent gains: FocalNet +0.39 dB / +0.001 SSIM, ConvIR +0.88 dB / +0.001 SSIM (best 38.83 dB), and SGDN +0.88 dB / +0.005 SSIM. We have incorporated this result into Table 1 of the revised paper.
>
> **Table E: Comparison of dehazing performance on SOTS-Outdoor dataset.**
> | Method                |   PSNR           |   SSIM          |
> |-----------------------|-----------------:|----------------:|
> | MSBDN                 | 33.48            | 0.982           |
> | FFA-Net               | 33.57            | 0.984           |
> | Dehamer               | 35.18            | 0.986           |
> | MB-Taylorformer       | 37.42            | 0.989           |
> | FocalNet              | 37.71            | 0.995           |
> | ConvIR                | 37.95            | 0.994           |
> | SGDN                  | 36.22            | 0.986           |
> | **HNDiff (FocalNet)** | **38.10 (+0.39)**| **0.996 (+0.001)** |
> | **HNDiff (ConvIR)**   | **38.83 (+0.88)**| **0.995 (+0.001)** |
> | **HNDiff (SGDN)**     | **37.10 (+0.88)**| **0.991 (+0.005)** |

---

> ### Author Response · Authors · 2025-11-25
>
> Dear reviewer,
>
> Thank you for the comments on our paper.
>
> We have submitted the responses to your comments. We would like to confirm whether our responses sufficiently addresses your concerns. Please let us know if you have additional questions so that we can address them during the discussion period.
>
> Thank you!
>
> Best,
>
> Authors

---

### Author Response · Authors · 2025-11-29
**Summary for Area Chair (Part 1)**

# Core Contribution of This Work:

- **A physically grounded diffusion framework.** We introduce HNDiff, a diffusion-based dehazing model that incorporates the Atmospheric Scattering Model (ASM) as an inductive bias.
- **A haze–noise diffusion formulation with adaptive noise scheduling.** The forward process jointly injects haze and noise, while the reverse process performs dehazing and denoising via two dedicated estimators, guided by a haze-aware scheduler that adjusts noise levels based on local haze density.
- **Comprehensive empirical validation.**  HNDiff consistently improves **three representative dehazing backbones** and achieves state-of-the-art performance across **seven benchmark datasets**.

# Review and Response Summary:
Across the reviews, several strengths of our work are consistently recognized, including strong physical grounding (reviewers ud8n, 96ea, JRmV, HBqs), rigorous theoretical derivations (ud8n, HBqs), practical relevance (96ea, JRmV), and strong empirical performance (96ea, JRmV, HBqs). Below, we summarize our responses for each reviewer, emphasizing how new experiments, clarifications, and additional evidence fully address their concerns.

## Reviewer ud8n
**W1: The impact of including a hazy image for the image encoder**

Using both hazy and clean images in Stage 1 allows the encoder to explore their correspondence. We conducted an ablation where the prior was generated with and without the hazy image. Including the hazy image yields a +0.62 dB improvement in Stage 1 PSNR on the benchmark dataset compared to the variant without the hazy image, demonstrating the effectiveness of our design choice.

**W2: Latent visualizations**

We updated Figure 5 to visualize latents via channel-wise averaging and clarified the resolution and downsampling steps, yielding more reliable and interpretable visualizations.

**W3-1: Three-stage training complexity**

The three-stage scheme follows recent latent diffusion restoration methods and has been widely validated as an effective and stable training strategy, as discussed in Section 3.3 of the paper.

**W3-2: Loss function in Stage 2**

Stage 2 uses only the latent-prior loss because intermediate haze residuals are unavailable; Gradients propagate through the entire diffusion trajectory, implicitly supervising the haze and noise estimators, as discussed in Section 3.3 of the paper.

**W4: Haze-aware noise scheduling**

While prior works only use haze cues as auxiliary guidance for denoising, our method integrates haze density directly into the forward diffusion process, resulting in a fundamentally different and novel noise scheduling mechanism.

**W5: Definition of HANS in Figure 2**

We addressed this issue by explicitly defining the Haze-aware Noise Scheduler (HANS) at the beginning of Section 3 before its first appearance in Figure 2.

**W6-1: Experiments on RTTS dataset**

We evaluated on RTTS and obtained consistent improvements across four strong backbones (FocalNet, ConvIR, SGDN, and RIDCP). In particular, on RIDCP, our method attains SOTA NR-IQA performance (FADE 0.944→0.417, NIMA 4.97→5.08, BRISQUE 17.29→16.09), while also boosting YOLOv3 detection mAP, demonstrating strong real-world effectiveness.

**W6-2: SOTS-outdoor dataset experiments**

We evaluated on SOTS-Outdoor as requested and obtained consistent PSNR gains across three backbones (FocalNet +0.39 dB, ConvIR +0.88 dB, and SGDN +0.88 dB ), demonstrating the robustness and effectiveness of our method.

**W6-3: Additional visual results**

We added full visual comparisons for all remaining baselines as requested. For the challenging cases in Figure 13, HNDiff actually reduces the severe artifacts seen in the baselines.

## Reviewer 96ea

**W1-1: Clarification of U-Net architecture**

We clarified that our U-Net follows the simplified design from Liu et al., as already stated in Section 4.1, and have added the detailed architecture description to the appendix.

**W1-2: Loss function for training the estimators**

We train both the haze and noise estimators using the latent-prior loss, as stated in Section 3.3 of the paper.

**W1-3: Dense haze scenarios**

We evaluated our method on the Dense-HAZE dataset, which contains extremely dense haze scenarios. HNDiff consistently improves all three backbones and achieves state-of-the-art performance (Table 1), demonstrating that the haze estimator remains robust in severe haze conditions.

**W2 & Q1-1: Non-uniform haze scenarios**

HNDiff naturally supports non-uniform haze through the per-pixel ASM formulation and our haze-aware scheduler, and our results on Table 1, all containing spatially varying haze, show consistent improvements over strong baselines.

**Q1-2: Avoiding confusion between image details and haze**

The haze estimator is explicitly designed to focus only on haze components, and our three-stage training further penalizes misinterpreting image details as haze, effectively preventing confusion between the two.

---

> ### Author Response · Authors · 2025-11-29
> **Summary for Area Chair (Part 2)**
>
> ## Reviewer JRmV
> **W1: Why this specific combination of haze and noise addition in Eq. (2) is more effective**
>
> Conventional diffusion adds only Gaussian noise and ignores how haze is physically formed. In Eq. (2), we inject haze as an ASM-guided mean shift of the Gaussian transitions, forcing the forward process to follow a physically grounded clean-to-hazy trajectory. The reverse step then jointly removes haze and noise, providing a stronger inductive bias than noise-only diffusion or deterministic haze estimation. This explanation has been added to Section 3.1.
>
> **W2: Hierarchical structure of the proposed method**
>
> We have restructured the beginning of Section 3 with a clear top-down hierarchy. The revised presentation first highlights our core contribution, the HNDiff diffusion framework, then explains the roles and relationships of its components (Haze-Noise Diffusion, HANS, and the Dehazing–Denoising Process). We also clarify how Latent HNDiff with FGM builds on the main framework for practical deployment.
>
> **Q1: Definition of HANS in Figure 2**
>
> We addressed this issue by explicitly defining the Haze-aware Noise Scheduler (HANS) at the beginning of Section 3 before its first appearance in Figure 2.
>
> **Q2: Dehazing features and performance within few steps**
>
> We clarify that the dehazing features correspond to haze residuals progressively removed during the iterative dehazing–denoising process. Our diffusion model acts as a latent-space prior generator that enhances the dehazing backbone, achieving high-quality restoration in only a few steps. This design also avoids extensive iterations typically required by image-space diffusion models.
>
> ## Reviewer HBqs
>
> This reviewer has already indicated that most of their concerns have been addressed, including those regarding the three-stage training difficulty, performance gains, convergence analysis, the interpretability of latent HNDiff, and the image-space variants of HNDiff. The remaining issues concern (i) whether our method can improve an existing ASM-based method on the RTTS dataset and (ii) adding more discussion of ASM-based methods in the paper, for which the reviewer has not yet responded. The reviewer also explicitly mentioned that he/she would raise the score if our method could address these concerns. Below, we summarize our responses related to these points.
>
> **(i) RTTS dataset experiment**
>
> We followed the reviewer’s suggestion by adopting RIDCP as a representative ASM-based backbone and evaluating HNDiff on RTTS. As shown in the updated results (Table 9 and Table 10 of the revised paper), HNDiff(RIDCP) consistently improves all NR-IQA metrics (FADE, NIMA, BRISQUE) and increases YOLOv3 mAP for every category, indicating that our prior substantially enhances existing ASM-based methods in real-world scenarios and yields cleaner dehazing results (Figure 22).
>
> **(ii) Discussion of ASM-based methods**
>
> Following the reviewer’s suggestion, we have added a discussion of prior ASM-based dehazing methods in Section 2.1 of the revised paper.

---

### Meta-Review · Area_Chair_mrCV · 2026-01-07

**Summary:**

This paper incorporates the Atmospheric Scattering Model (ASM) into the diffusion process so that the diffusion process follows physical principles. Specifically, the paper proposes a Haze-Aware Noise Scheduler (HANS), which adds different noise levels to regions with different haze levels, improving the generation ability in heavily hazy regions. It then introduces Latent HNDiff, which performs the diffusion process in the latent space to improve efficiency.

The AC also reads the paper through without looking at the reviews and provide the below pros and cons.

Strengths:
1. Applying different noise levels to regions with different haze levels during the forward diffusion process is a reasonable design.
2. Defining the forward process as a deterministic mean shift allows the joint denoising and dehazing process to be explicitly solved.
3. The experimental results are promising.

Weaknesses:
1. Figure 2 is difficult to understand. It does not clearly illustrate the two-stage training process, nor does it clearly show how the losses are computed in each stage.
2. In the first stage, an image encoder is trained to extract the “ground truth prior.” The meaning of “ground truth prior” is unclear.
3. The main motivation is to integrate the physical model of hazy images into the diffusion process. However, the whole diffusion process is conducted in the latent space. There is no concrete ground why the physical laws defined in the image space shall hold in the latent space.

**Reviewer Concerns:**

- The three-stage design is complicated;
- There are distinct components in the proposed framework, but there is no clear structure or relations between them;
- Missing real-scenario experiments;
- The gain of +0.5dB is relatively small.

**Reviewer Scores:**

Reviewer ud8n gave 4 before the rebuttal. The reviewer's concerns on the three-stage design and real-scenario experiments shall be partially solved by the rebuttal. In the rebuttal, the authors mentioned that related methods also using three-stage designs and additional experiments on RTTS dataset. The AC thinks the rebuttal partially solves the concerns but not entirely, as there are also strong methods without the three-stage design and the experiments on RTTS shall be put in the first manuscript but not in rebuttal. The rebuttal has limited space and cannot discuss the real-scenario experiments throughoutly.

Reviewer 96ea gave 8 before the rebuttal and is likely to maintain 8 after the rebuttal.

Reviewer JRmV gave 4 before the rebuttal. The main concerns are that there is no clear, intuitive explanation of the proposed framework and there is no clear motivation and relations between the proposed component. The rebuttal might not able to fully solve the reviewer's concerns. The AC reads the paper and has the same feeling. It is more like a combination of tricks here and there without sufficient theoritical ground.

Reviewer HBqs gave 4 before the rebuttal. His/her main concern are 1) the three-stage design is complicated, 2) the method is a combination of ASM-based method + latent diffusion model, 3) there is a lack of real scenario experiments. The rebuttal might partially solved the concerns but might not totally solve them. Please see the first paragraph.

---

### Decision · Program_Chairs · 2026-01-26

Reject